# Metformin: The Winding Path from Understanding Its Molecular Mechanisms to Proving Therapeutic Benefits in Neurodegenerative Disorders

**DOI:** 10.3390/ph16121714

**Published:** 2023-12-11

**Authors:** Laura Mihaela Isop, Andrea Elena Neculau, Radu Dan Necula, Cristian Kakucs, Marius Alexandru Moga, Lorena Dima

**Affiliations:** 1Department of Fundamental, Prophylactic and Clinical Sciences, Faculty of Medicine, Transilvania University of Brasov, 500036 Brașov, Romania; laura.isop@unitbv.ro (L.M.I.);; 2Department of Medical and Surgical Specialties, Faculty of Medicine, Transilvania University of Brasov, 500036 Brașov, Romania

**Keywords:** metformin, diabetes, neurodegenerative, neuroprotection, antiaging, cognitive impairment, Alzheimer, Parkinson disease, epilepsy

## Abstract

Metformin, a widely prescribed medication for type 2 diabetes, has garnered increasing attention for its potential neuroprotective properties due to the growing demand for treatments for Alzheimer’s, Parkinson’s, and motor neuron diseases. This review synthesizes experimental and clinical studies on metformin’s mechanisms of action and potential therapeutic benefits for neurodegenerative disorders. A comprehensive search of electronic databases, including PubMed, MEDLINE, Embase, and Cochrane library, focused on key phrases such as “metformin”, “neuroprotection”, and “neurodegenerative diseases”, with data up to September 2023. Recent research on metformin’s glucoregulatory mechanisms reveals new molecular targets, including the activation of the LKB1–AMPK signaling pathway, which is crucial for chronic administration of metformin. The pleiotropic impact may involve other stress kinases that are acutely activated. The precise role of respiratory chain complexes (I and IV), of the mitochondrial targets, or of the lysosomes in metformin effects remains to be established by further research. Research on extrahepatic targets like the gut and microbiota, as well as its antioxidant and immunomodulatory properties, is crucial for understanding neurodegenerative disorders. Experimental data on animal models shows promising results, but clinical studies are inconclusive. Understanding the molecular targets and mechanisms of its effects could help design clinical trials to explore and, hopefully, prove its therapeutic effects in neurodegenerative conditions.

## 1. Introduction

Even after more than six decades since its initial introduction, metformin (1,1-dimethylbiguanide hydrochloride) continues to hold its position as the most frequently prescribed oral medication for reducing glucose levels in the treatment of type 2 diabetes (T2D) [1]. The World Health Organization (WHO), along with sulfonylureas (SU), and insulin, added metformin to its list of essential medications in 2011. This ruling highlighted the significant part metformin plays in the global care of people with T2D [2].

The UK Prospective Diabetes Study (UKPDS), a landmark study, significantly altered the perception of long-term advantages of glycemic control in diabetic patients. Over 5000 patients were randomly assigned to glucose-lowering therapy groups, including metformin, sulfonylurea, or insulin. The results showed metformin’s positive effects, reducing the risk of diabetes-related mortality and fewer hypoglycemia episodes compared to alternative pharmacological therapies [3]. Metformin was then suggested as a first-line medication for the treatment of type 2 diabetes. Several clinical studies have found that metformin is a successful pharmaceutical adjunct for mitigating the metabolic dysfunction induced by antipsychotic medications, commonly referred to as antipsychotic-induced metabolic dysfunction [4,5,6]. In the UK, the use of metformin increased from 55.4% in 2000 to 83.6% in 2013 [7], while in the US, it increased from 60% in 2005 to 77% in 2016 [8].

There is also evidence that metformin can help with diabetes prevention, gestational diabetes, and polycystic ovarian syndrome (PCOS) [9,10]. Experimental evidence suggests that metformin could potentially offer anti-thrombotic and anti-atherosclerotic benefits [11,12,13], positively impact the gut microbiome and immune function [14,15], play a role in enhancing endothelial cell function, and contribute to reducing fasting hyperinsulinemia and inflammation [11,12,13]. 

Given its pleiotropic effects, proved in preclinical studies, research has been developed into the area of neuroprotection, with possible benefits for neurodegenerative and some neuropsychiatric conditions. Metformin’s favorable benefits extend to neurological illnesses; metformin has been shown to improve the pathophysiology of Alzheimer’s, Parkinson’s, and Huntington’s diseases, as well as multiple sclerosis [16].

Neurodegenerative diseases, such as Alzheimer’s, Parkinson’s, and motor neuron diseases, are becoming increasingly prevalent worldwide, particularly among the elderly population [17]. These disorders involve the gradual breakdown and impairment of neurons in the brain and spinal cord, leading to a decline in cognitive capabilities, motor functions, and overall neurological performance [16]. These diseases are characterized by a decline in both structure and functionality of the impacted neurons [16].

Between 2016 and 2017, neurodegenerative illnesses affected 4.7 to 6.0 million people in the US, causing 272,644 deaths and 3,011,484 disability-adjusted life years [17,18,19,20]. The number of common dementia cases grew by 117% between 1990 and 2016, rising from 202 million (174–235) in 1990 to 438 million in 2016 [20,21]. This is concerning as there is no cure or treatment for dementia, making the situation even more challenging [22]. Clinical trials for potential treatments are extremely low, with a ratio of failed trials to successful ones exceeding 100:1, compared to the pharmaceutical industry average of 14.6:1. This highlights the difficulty in finding effective treatments for dementia [23,24]. The prevalence of dementia doubles every five years over the age of 50, and the absence of a cure makes it crucial to examine risk factors closely [19]. The Lancet Commission Report suggests that 35% of the dementia burden could be alleviated by addressing modifiable risk factors like hearing loss, education, smoking habits, depression, lack of physical activity, social isolation, diabetes, and obesity [25]. The absence of curative treatments puts a significant burden on individuals, caregivers, and healthcare systems. Understanding the mechanisms behind dementia and identifying effective interventions is crucial for reducing the disease’s burden. Although there is a great deal of research in the area of small molecules, genetic, and biologic therapies targeting factors involved in the pathophysiology of AD (amyloid β (Aβ), tau protein, or innate immune activation states) in the hope of finding disease-modifying treatments, there are still many open questions, and the burden on patients and families has not yet been overcome [26].

Neurodegenerative diseases often require early detection and treatment, but current methods are limited. Current treatments focus on symptom management, but do not effectively slow disease progression [27,28]. The blood–brain barrier poses a challenge for drug delivery, and the heterogeneity of diseases complicates treatment approaches. The high cost of research and development further complicates the process, with clinical trials often failing to bring effective treatments to market [27]. Therefore, there is a need for disease-modifying treatments to slow or stop disease progression [26].

Paralleling the search for innovative therapy to change AD into a preventable or curable one, the concept of repurposing existing drugs is gaining momentum in research. There is growing interest in exploring the potential of diabetes drugs, specifically metformin, for the treatment of neurological disorders. It has been suggested that these drugs may offer new therapeutic avenues for the management of such conditions. Metformin has shown promise in both clinical and animal studies by exerting possible neuroprotective effects in various neurological disorders [27,28]. While more research is needed to fully understand the precise mechanisms and efficacy of metformin in treating neurological disorders, these findings present exciting possibilities for the field. Repurposing drugs like metformin can potentially expedite the development of treatments for neurological conditions, leveraging existing knowledge and safety profiles to address unmet medical needs [29,30,31,32]. This approach holds great potential for advancing therapeutic options and improving outcomes for individuals with these disorders.

Diabetes and pre-diabetes may contribute to dementia development due to a complex link between glucose metabolism and cognitive impairment [33]. Factors include mitochondrial dysfunction, oxidative stress, protein glycation, aldose reductase activity, and intracellular signal transduction pathways. Poor glycemic management and hypoglycemia increase the risk of dementia and cognitive decline, with the impact of glucose-lowering medications on cognition potentially mediated by hyperglycemia or hypoglycemia [34,35]. Furthermore, it was observed that the occurrence of cancer, in general, was reduced in individuals undergoing metformin treatment [36]. Building upon these encouraging findings, a clinical trial called TAME (Targeting Aging with Metformin; http://www.afar.org/natgeo, accessed on 30 August 2023) aims to investigate whether metformin has the potential to extend the onset of age-associated illnesses and ailments, encompassing cancer, cardiovascular disorders, and AD.

In the context of extensive research in recent decades, the purpose of this narrative review was to provide an overview of the current understanding of metformin’s mechanisms of action derived from in vivo and in vitro studies, highlighting specific targets that have relevance to the treatment of various neurodegenerative conditions, as well as the available level of evidence of its potential therapeutic benefits derived from clinical studies.

## 2. Metformin Pharmacology

The UKPDS, a 20-year study, found that metformin usage significantly increased among individuals with type 2 diabetes, highlighting its cardiovascular benefits [3]. Metformin’s anti-diabetic effects are thought to be predominantly mediated by hepatic activities, and its effects on lipid and glucose metabolism are thought to have more significant therapeutic benefits. Metformin’s antihyperglycemic actions in diabetic individuals were shown to be primarily caused by inhibiting hepatic gluconeogenesis [37]. Additionally, metformin increases insulin-stimulated systemic glucose disposal—mainly in skeletal muscle [38,39,40]—inhibits lipid synthesis, and stimulates fatty acid oxidation, lowering free fatty acid levels. Apart from peripheral glucose disposal, metformin has other extra-hepatic sites of action, the role of the metformin action in the gut currently being extensively investigated [41]. The clinical advantages of metformin as an anti-hyperglycemic medication are significantly influenced by the intestine. This influence is achieved through the modulation of glucagon-like peptide 1 (GLP-1) levels, which occurs through a pathway mediated by duodenal AMPK (adenosine monophosphate-activated protein kinase). Additionally, metformin’s effects are also influenced by changes in composition of the gut microbiota, further contributing to its beneficial outcomes [42,43,44]. 

### 2.1. Pharmacokinetic Properties of Metformin

Metformin is a hydrophilic compound found in the form of monoprotonated cation over the entire range of physiological pH. Given the positive charge, its passive diffusion through cell membranes is limited and its absorption, distribution, and excretion strongly rely on specific transporters: organic cation transporters (OCTs), multidrug and toxin extruders (MATEs), and possibly the plasma membrane monoamine transporter (PMAT) [38]. OCT transporters in the SLC22A family generally mediate the entry of organic cations into cells. Organic cation transporters 1 (OCT1/SCL22A1), organic cation transporters 3 (OCT3/ SCL22A3), and new organic cation transporters 1 (OCTN1) transport metformin, which is mostly absorbed and distributed in intestinal epithelial cells and hepatocytes [45,46,47]. Other transporters, such as the carnitine/organic cation transporter 2 (OCTN2), plasma membrane monoamine transporter (PMAT), thiamine transporter 2 (THTR2), or serotonin transporter (SERT) have also been involved in intestinal absorption of metformin [48]. The apical membrane transporters facilitate enterocyte influx, while basolateral membrane transporters release substances into the blood. Metformin absorption is saturable, possibly due to lack of efflux intestinal transporters, causing higher concentrations in the gut [49].

OCT1 and OCT3 are involved in metformin uptake into skeletal muscle, another important site of metformin action [50]. OCT2, located on the basolateral membrane of renal proximal tubular epithelial cells is involved in active secretion of metformin by influx within the cells, while MATE 1 and 2, located at the apical pole of the membrane eliminates the drug into urine [50]. In humans, metformin accumulates to substantially larger quantities in the liver, kidney, and small intestine, with minimal uptake peripherally [51]. The tissue distribution is consistent with the expression of the transporters [52]. However, 11C-metformin PET–CT, used to assess metformin biodistribution in humans, cannot assess the distribution at the subcellular level, i.e., accumulation within specific organelles [51]. An overview of transporters relevant to metformin pharmacokinetics is presented in Figure 1.

Although the precise mechanisms by which metformin is transported in the central nervous system are still unclear, recent in vivo and in vitro studies have shown that it can pass the blood–brain barrier and can work by activating certain neurons and neuroglia to provide its desired effects [53,54]. Moreover, metformin transporting has been linked to the equilibrative nucleoside transporter (ENT) family, of which PMAT is a member, and which is present in many bodily tissues, including the brain, central nervous system [44], the blood–brain barrier and brain express OCT3 [55]. Research on the metformin transporter in the neurological system is currently ongoing, and further investigation is required to clarify the precise processes at play. Nevertheless, the fact that metformin may penetrate the blood–brain barrier and operate neurologically points to possible therapeutic benefits for the central nervous system [53,54].

An approximate 55% bioavailability is produced by incomplete transporter-mediated absorption from the upper intestine [56,57]. The bioavailability after oral administration has been found to vary between 40 and 60%. It is predominantly absorbed in the upper small intestine, systemically distributed within 6 h following absorption, and reaches a peak concentration at about 3 h [38]. Metformin is eliminated unchanged in urine, primarily through passive glomerular filtration and active transportation facilitated by OCT 2 and MATEs [56,57,58].

Although these transporters play significant roles in processes of absorption, distribution and elimination of metformin, no substantial association between glycemic response to metformin and genetic variability of their corresponding encoding genes have been found in patients with T2DM [59]. Rather there are pharmacogenetic studies on genes involved in the metformin pharmacodynamics that revealed influences on clinical response.

### 2.2. Mechanisms Involved in Glucoregulation

Metformin inhibits gluconeogenesis, the process by which the liver synthesizes glucose from non-carbohydrate sources such as lactate, glycerol, and amino acids, which consequently lowers glucose synthesis in the liver. Metformin stimulates AMPK, a cellular regulator of lipid and glucose metabolism [60,61,62]. AMPK is activated by metabolic stress and inhibits hepatic gluconeogenesis while also improving insulin sensitivity, muscular glucose uptake, and fatty acid oxidation. 

In addition to these, AMPK-mediated inhibitory phosphorylation inhibits/limits 3-hydroxy-3-methylglutaryl coenzyme A (HMG-CoA) reductase, an enzyme that is a key rate-limiting protein in cholesterol biosynthesis (Figure 2) [63]. By phosphorylating and inactivating enzymes involved in the gluconeogenic pathway, such as phosphoenolpyruvate carboxykinase (PEPCK) and glucose-6-phosphatase (G6Pase), activated AMPK suppresses hepatic gluconeogenesis. The inactivation of these enzymes inhibits the conversion of substrates (such as lactate, amino acids, and glycerol) into glucose.

### 2.3. AMPK Signaling Pathway

AMPK is activated by a low energy status, signaled by raising the AMP to ATP (adenosine monophosphate to adenosine triphosphate) or ADP (adenosine diphosphate) to ATP ratios (within cells and is caused by an 8% decrease in ATP and a four-fold rise in AMP) [64]. In consequence, metabolic adaptation to restore the energy level is facilitated by suppressing anabolic ATP-consuming processes such as protein synthesis and fatty acid synthesis while increasing catabolic ATP-generating routes such as glycolysis, glucose absorption, and fatty acid oxidation [65,66]. Apart from this AMP/ADP-dependent activation, the ‘canonical’ pathways, which is in line with the classical role of AMPK as an energy sensor, AMP-independent activation mechanisms of AMPK, non-canonical pathways by which AMPK senses the level of glucose, glycogen or fatty acids, or damage to lysosomes and nuclear DNA have been described [67].

Metformin can activate AMPK through both AMP-dependent and AMP-independent pathways depending on its concentration and target organelles like mitochondria or lysosomes [49]. Figure 3 shows a schematic representation of the two pathways.

Activation of the LKB1(Liver kinase B1)–AMPK signaling pathway was considered, for a long time, the main mechanism explaining the metformin therapeutic effects on glucose metabolism (Figure 2) [68]. An AMPK-dependent model of metformin-induced inhibition of gluconeogenesis has been proposed, according to which activation of the LKB1–AMPK signaling pathway by metformin would play an important role in inhibiting the expression of gluconeogenic gene transcription (Pck1 and G6pc) via the phosphorylation and cytoplasmic sequestration of the transcriptional cofactor cAMP- response element- binding protein (CREB)-regulated transcription co-activator 2 CRTC2 [68]. The model has been challenged by later studies providing substantial evidence that AMPK-independent mechanisms are also involved in metformin actions. The acute glucose-lowering effect of metformin was maintained in genetic loss-of-function experiments, such as liver-specific, intestine-specific, and skeletal muscle-specific AMPK-deficient mice [49,69,70]. Other kinases might be involved in metformin’s acute effects [49]. However, AMPK activation plays an important role in chronic metformin administration [49,71,72].

Zhang et al. discovered an alternate mechanism for AMPK activation using minimal metformin concentrations, focusing on the relocation of AXIN, a scaffold protein, and LKB1 to the lysosomal surface [73,74,75]. This mechanism, which does not rely on AMP, could be triggered by metformin at low doses. Chen et al. found metformin suppresses mTORC1 (mammalian target of rapamycin complex 1) and activates AMPK through the lysosomal pathway, prolonging *C. elegans* lifespan and mitigating aging fitness decline linked to lysosomal pathway components [76].

Ma et al. showed that a low dose of metformin interferes with the vacuolar H+-ATPase in the lysosomal proton pump (v-ATPase), raising doubts about its mechanism of action in real in vivo scenarios. The interaction between metformin and PEN2 forms a complex involving ATP6AP1, inhibiting v-ATPase function, and activating AMPK, without impacting cellular AMP levels [72]. Metformin, when administered at higher concentrations, can bypass PEN2–ATP6AP1 signaling to initiate AMPK activation on the lysosome surface [70,72]. By blocking acetyl-CoA carboxylase (ACC) and increasing the release of GLP1 in the gut, metformin-activated AMPK from lysosomes reduces fat accumulation in the liver and lowers blood glucose levels [49]. However, its effectiveness in humans remains uncertain due to its limited effectiveness in non-alcoholic fatty liver disease [77,78]. Remarkably, in primary hepatocytes exposed to small doses of metformin, the PEN2-ATP6AP1 pathway activates AMPK that is in lysosomes while leaving other pools of AMPK, such as those found in the endoplasmic reticulum and mitochondria, unchanged [72]. 

In summary, the most recent discoveries about the glucoregulatory mechanisms of action of metformin have revealed new plausible variables, adding a few more complexities and ambiguities. Studies finding that the phosphorylation events may occur without the involvement of LKB1–AMPK signaling [49] suggest that the pleiotropic impact of metformin may involve other stress kinases that are acutely activated. However, AMPK activation plays an important role in chronic metformin administration. In addition, examining the lysosome as an extra target organelle offers understanding of new ways by which metformin acts via the AMPK signaling pathway.

### 2.4. Mitochondrial Respiratory Chain Complexes

Mitochondrial complex I, recognized as NADH (nicotinamide adenine dinucleotide) dehydrogenase or NADH: ubiquinone oxidoreductase, constitutes an integral element of the oxidative phosphorylation (OXPHOS) machinery [79]. It stands as an extensively investigated candidate for metformin’s potential targets. The suppressive influence of metformin on complex I was initially documented in the liver, and subsequently, this effect has been confirmed in various other tissue types and cellular models [36,80,81].

Metformin’s positive charge at pH 7.4 (where nearly all of it exists in an ionized form in the blood) makes it prone to concentrate in negatively charged organelles like mitochondria [56,82,83]. Metformin exhibits a capacity to specifically inhibit mitochondrial complex 1, thus targeting mitochondria and mitochondrial accumulation is often considered the primary target of metformin [82,84]. Within the mitochondrial respiratory chain, complex 1 stands as the largest protein entity, comprising no fewer than 45 subunits. Its significance is underscored by its role in influencing the initiation and progression of cancer [85]. 

Mitochondria play a critical role in energy metabolism. Metformin’s action on mitochondrial complex I allows it to modulate this process, leading to a decrease in the relative energy charge within cells (Figure 2). As a result, metformin assumes a pivotal role in maintaining cellular energy balance [86]. These effects have a number of impacts, including depolarization of the mitochondrial membrane potential, increased ratios of AMP/ATP accompanied by an increase in cellular redox potential and the NADH/NAD^+^ ratio [58,82]. As the gluconeogenetic processes are dependent on ATP level, gluconeogenesis is inhibited. Meanwhile, an increase in AMP will inhibit the activity of adenilacyclase and fructose-1-6-bisphosphatase (FBP1), both involved in gluconeogenesis. The extent of gluconeogenesis inhibition correlates with the degree of respiratory chain suppression, indicating that metformin-induced cellular energy depletion leads to an incomplete ATP flux necessary for initiating gluconeogenesis in the liver [58,83].

A 2023 study using cryo-electron microscopy and enzyme kinetics identified three potential binding sites for biguanides on complex I protein subunits. The most significant inhibitory site is in the amphipathic region of the quinone-binding channel. It has been observed and documented before that when a biguanide binds to this location, it effectively prevents the reactivation of the enzymatic inactive state [87,88].

Studies in 2000 revealed that metformin specifically suppresses mitochondrial complex I while leaving complexes II, III, and IV unaffected [58,80,81]. In contrast to the proposed hypothesis, a study published in 2022 conducted by LaMoia et al. presents evidence indicating that the inhibition of complex I activity, both in vitro and in vivo, does not lead to a reduction in plasma glucose concentrations or the inhibition of hepatic gluconeogenesis [89]. Instead, the study goes on to demonstrate that metformin inhibits complex IV activity at concentrations relevant to clinical applications (Figure 2) [89]. This inhibition of complex IV activity subsequently triggers the inhibition of glycerol-3-phosphate dehydrogenase activity, resulting in heightened cytosolic redox levels and a specific suppression of hepatic gluconeogenesis stemming from glycerol both in vitro and in vivo. 

### 2.5. mGPDH and Cytosolic Redox State

Additionally, metformin decreases blood glucose via a redox mechanism [78] including a 30–50% suppression of mitochondrial glycerol phosphate dehydrogenase (mGPD) [79], which inhibits gluconeogenesis in the liver. mGPD is the rate-limiting enzyme of the cytosolic redox NAD^+^/NADH shuttle and the glycerol phosphate shuttle (Figure 2) [80].

Positioned within the outer layer of the inner mitochondrial membrane, mGPDH plays a crucial role. Metformin has been identified as directly interacting with this enzyme, leading to its functional inhibition [90,91]. This direct interaction results in a noteworthy decrease in mitochondrial respiration of glycerol-3-phosphate (G-3-P) and its conversion into dihydroxyacetone phosphate (DHAP). mGPDH functions as a key player in the glycerophosphate shuttle. Within the context of NAD^+^ regeneration, the glycerophosphate shuttle assumes the responsibility of reducing NADH within the mitochondrial matrix [90,92].

The glycerophosphate shuttle and malate-aspartic acid shuttle facilitate the transfer of reduced NADH into mitochondria, enabling ATP production and replenishing cytoplasmic NAD^+^ [93]. Metformin inhibits mGPDH, reducing liver glucose production in mice. Downregulating GPD2 mimics the glucose lowering effect, but in GPD2-deficient mice, metformin fails to lower blood glucose levels [94].

Metformin, when used in clinical settings, increases the liver’s cytosolic NADH to NAD^+^ ratio without significant changes in intracellular ATP levels [90,95,96]. The increased NADH to NAD^+^ ratio reduces glucose production from reduced gluconeogenic substrates like lactate and glycerol, while glucose production from oxidized substrates remains unaffected [49].

The proposed model of metformin-triggered mGPDH inhibition faces challenges, as inhibiting the glycerol-phosphate shuttle may not be enough to significantly decrease gluconeogenesis in the liver [92]. Despite the commonly demonstrated impact of low metformin doses on increasing the cytosolic NADH to NAD^+^ ratio, not all studies have consistently shown a reduction in glucose production from lactate or a direct inhibition of mGPDH activity upon metformin treatment [96,97,98].

### 2.6. Interaction of Metformin Treatment, Metabolites, and Gut Microbiota

The gut microbiota, comprising approximately 1014 bacteria from nine phyla, plays a crucial role in human growth, development, physiological processes, and disease states, aiding the body in nutrient extraction and metabolic regulation [99,100,101].

Even though metformin’s effect on gut microbiota in T2DM patients differs significantly by ethnicity, the enrichment of Bacteroides operational taxonomic units and the reduction in Faecalibacterium units are the therapy’s defining features, which are constant across ethnicities [102,103]. In comparison to healthy people and diabetes patients not receiving metformin, diabetes patients receiving metformin showed greater abundance of mucin-degrading bacteria and GM bacteria that produce short-chain fatty acids (SCFAs) [104,105]. By increasing the abundance of Lactobacillus and altering the glucose-SGLT1-sensing glucoregulatory system, metformin also had antihyperglycemic effects [106].

Metformin’s use is linked to modification of the bile acid pool, reducing the prevalence of Bacteroides fragilis and its bile salt hydrolase activity [107]. Increased levels of glycoursodeoxycholic acid enhance glucose metabolism balance by inhibiting intestine farnesoid X receptor signaling through an AMPK-independent mechanism [108].

## 3. Mechanisms Involved in Central Nervous System Functions and Neuroprotection

The complex regulation of the central nervous system’s activities in the context of neurodegeneration is vulnerable to a variety of complex processes, each of which contributes in a different but related way to the development of crippling disorders. The mechanisms that are involved—such as inflammation, oxidative damage, mitochondrial dysfunction, impaired autophagy, aberrant protein phosphorylation, and disruption of insulin signaling and glucose homeostasis —become crucial factors in determining the course of neuronal health within this complex web. Figure 4 provides an outline of the mechanisms involved in neurodegeneration.

Molecular events play a significant role in neurodegenerative disorders, marked by an escalation of inflammatory cytokines orchestrated by microglia and astrocytes, inducing a pro-inflammatory state in the central nervous system. Elevated levels of ROS give rise to oxidative stress, disrupting the delicate balance of redox homeostasis. NF-κB signaling amplifies the inflammatory response, fostering the progression of neurodegeneration. The decline in antioxidants, crucial guardians against oxidative stress, heightens vulnerability. Furthermore, alterations in insulin signaling and glucose homeostasis contribute to metabolic dysregulation, a critical factor in neurodegenerative processes. Additionally, aging shares almost parallel mechanisms with neurodegeneration. The impact of metformin across these pathways suggests its potential to mitigate these intricate events.

### 3.1. Insulin Signaling

Insulin has a significant function in the processes of learning and memory. It achieves this by interacting with insulin receptors, which influence synaptic plasticity, the functions of excitatory and inhibitory receptors, and the modification of gene expression that is crucial for the consolidation of long-term memories [109,110].

Insulin receptors are widely found in the brain, primarily in the cortex and hippocampus. The insulin receptor belongs to a group of tyrosine kinase receptors. When islet molecules bind to the IR, it triggers the phosphorylation of tyrosine residues in the IR dimer. This leads to the phosphorylation of insulin receptor substrates (IRS), initiating a cascade of events that activate downstream pathways such as phosphatidylinositol 3-kinase (PI3K)/protein kinase B (Akt) and mitogen-activated protein kinase (MAPK). These activated pathways produce various effects downstream [109,111,112,113].

Moreover, glutamate-induced neurotoxicity may be attributed to inactivating MAPK and PI3K signaling [114]. By maintaining MAPK and PI3K active, metformin protects neurons from glutamate-induced neurotoxicity. In A-treated SH-SY5Y cells, it was shown that metformin therapy directly decreased the amounts of Ca(^2+^), reactive oxygen species (ROS), and extracellular and internal glutamate [115,116,117].

In the hippocampus and cerebral cortex of intracerebroventricular injection of streptozotocin-induced AD mice, phosphorylation of the insulin receptor, which is the initial reaction that occurs after the receptor is engaged, experienced a notable reduction after intranasal treatment with metformin, according to Kazkayasi et al. [118]. The effects of intranasal metformin on pIR and pAkt imply that it restores the damaged insulin signaling pathway by enhancing the insulin sensitivity of neurons in any area of the brain [118].

Also, metformin can inhibit the expression of insulin and IGF-1 receptors, as well as the phosphorylation of insulin receptors and IRS1, contributing to its effects on glucose metabolism and insulin signaling [119,120]. 

### 3.2. Antioxidant Mechanism of Metformin

Reactive oxygen species (ROS) are highly reactive molecules containing oxygen that are produced as natural by-products of various cellular processes, including aerobic cell energy metabolism and catecholamine neurotransmitter metabolism. Neurons are particularly vulnerable to oxidative stress due to their high oxygen consumption and high content of polyunsaturated fatty acids, which are susceptible to ROS-induced damage. The delicate balance between ROS generation and elimination is crucial for neuronal health and function. Disruptions to this balance can contribute to various neurodegenerative diseases and other neurological disorders; as a consequence, the antioxidant system is critical for mitigating the effects of ROS overproduction [121].

Metformin therapy significantly boosted endogenous antioxidant enzymes such as superoxide dismutase (SOD) and glutathione (GSH) levels in brain tissue, according to one study [16]. Also, metformin exerts its impact on disease-related pathways by obstructing complex I in the mitochondria, resulting in a deceleration of oxidative phosphorylation and a reduction in gluconeogenesis. This mechanism aids in alleviating the oxidative stress on neurons by minimizing the utilization of NADH [16]. Moreover, research conducted by An et al. revealed that metformin exhibits antioxidant properties in human umbilical vein endothelial cells through an AMPK-dependent pathway [122].

Metformin can indeed activate the forkhead box O 3 (FOXO3) protein, a gene associated with healthy aging in centenarians, a lower prevalence of cardiovascular events and a better outcome in inflammatory disorders [123,124], leading to various cellular effects, including the reduction in ROS and reactive nitrogen species (RNS) levels in immune cells [125].

Several pathways are involved in oxidative stress in individuals with DM. Metformin had a notable impact on diabetic GK rats’ brains, leading to significant reductions in thiobarbituric acid reactive substances (TBARS), malondialdehyde (MDA), and carbonyl groups [126,127]. These findings indicate that metformin exhibited antioxidant effects in the brain cells of GK diabetic rats, resulting in decreased levels of H_2_O_2_ and oxidized form of glutathione (GSSG), along with increased levels of reduced form of glutathione (GSH) and vitamin E [126,127]. Additionally, metformin activated glutathione peroxidase, glutathione reductase, and manganese superoxide dismutase enzymes (MnSOD), further contributing to its antioxidant benefits. Notably, MDA and TBARS, byproducts of lipid peroxidation, act as free radicals within brain cells, triggering processes associated with neurodegeneration [58,85].

### 3.3. Anti-Inflammatory Mechanisms of Metformin

Neuroinflammation is a common occurrence in different central nervous system disorders and plays a significant role in cognitive impairment. Research has indicated that neuroinflammation contributes to the development of neurodegenerative diseases. When microglia cells become abnormally activated, it can result in immune dysfunction and inflammation, causing the release of inflammatory substances that harm nearby healthy nerve tissue. Over time, this process leads to the gradual degeneration and death of neurons [83,128]. Chronic neuroinflammation, driven by the persistent abnormal activation of microglia, plays a significant role in the development and progression of neurodegenerative diseases such as AD, PD, and HD. Prolonged inflammation can lead to the destruction of neurons and synaptic structures in the brain [129]. The findings of a study by Rabieipoor suggest that metformin therapy might reduce neuroinflammation by reducing astrocyte activity (astrogliosis) and microglial activation [130].

Studies have indicated that metformin can suppress NF-κB (nuclear factor kappa-light-chain-enhancer of activated B cells) signaling and reduce the production of pro-inflammatory cytokines in various cell types [129,131]. NF (nuclear factor) has been shown to have a protective role in the brain, as it regulates the dephosphorylation of the Tau protein through miR-141-mediated modulation of protein phosphatase 2A (PP2A) [132]. Also, Xu et al. concluded that metformin’s action involves inhibiting the acetylation of histone H3, leading to the inactivation of NF-κB. This inactivation subsequently results in the suppression of pro-inflammatory gene expression [133].

Studies also have suggested that in the brains of APP/PS1 mice, metformin has the notable ability to substantially reduce the concentrations of proinflammatory cytokines including tumor necrosis factor-alpha (TNF-α), interleukin 1 beta (IL-1β), and IL-6. This effect is likely due to the observed activation of the AMPK/P65 NF-κB signaling pathways in the hippocampus [134,135]. In a five-year follow-up study, it was observed that metformin monotherapy in older diabetic individuals resulted in reduced levels of circulating proinflammatory cytokines [136]. This reduction in proinflammatory cytokines was associated with a decreased risk of mortality in this population [136].

Metformin has also been advocated as an additional medication in the treatment of typical inflammatory and immune-mediated diseases. A 12-week therapy with metformin 2550 mg/day resulted in decreased blood levels of high-sensitive CRP compared to a placebo in a study of individuals receiving glucocorticoid treatment for diverse chronic inflammatory disorders. Furthermore, carbohydrate-challenged TNF- levels increased considerably in the placebo-treated group but not in the metformin-treated group [137].

Metformin reduces hyperglycemia stress and inflammation by disrupting the interaction between Caveolin1 and AMPK [138]. Also, in astrocytes treated with high glucose levels, metformin therapy reduces inflammation and inhibits the elevated endoplasmic reticulum stress [138]. Likewise, a study investigating microglial cells stimulated with LPS (lipopolysaccharide) to mimic an ischemia/reperfusion (I/R) event and treated with metformin, revealed that these cells produced elevated levels of the anti-inflammatory cytokine interleukin-10 (IL-10) compared to the control group [139].

### 3.4. Neurogenesis

In one particular study, it was demonstrated that continuous administration of metformin can lead to the growth of new neurons in the dentate gyrus area, enhance brain energy metabolism, and improve learning and memory capabilities in obese mice by influencing the makeup of the gut microbiota [140]. In a study focused on investigating the long-term neuroprotective effects of metformin after hypoxic ischemia injury, it was observed that metformin administration resulted in reduced neuronal degeneration in the CA1 region of the hippocampal area [141]. Indeed, another study provided confirmation that metformin has the capability to stimulate the proliferation and differentiation of neuroblasts in the hippocampal region of the brain [142]. 

### 3.5. Autophagy-Inducing Mechanism of Metformin

In neurodegenerative diseases, the process of autophagy, which is responsible for the removal and recycling of damaged cellular components, is frequently disrupted or dysregulated. This dysregulation of autophagy can lead to abnormal accumulation of proteins and dysfunction of organelles within neurons [143,144].

Metformin treatment has been shown to boost the process of autophagic clearance of hyperphosphorylated tau in HT22 cells exposed to high glucose conditions and in the brains of db/db mice. This enhancement of autophagy occurs through the activation of AMP-activated protein kinase (AMPK) which reduced cognitive impairment in db/db mice [145]. Metformin has been shown to improve the Th17 inflammaging profile by increasing autophagy and restoring normal mitochondrial function, concluded Bharath et al. [11]. 

### 3.6. The Putative Relationship between Metformin and Aging

Metformin acts on multiple pathways associated with stem cell exhaustion, which provides a compelling reason to explore its potential role in addressing age-related stem cell exhaustion. By targeting these pathways, metformin may offer a promising avenue to better understanding and potentially mitigating the decline in stem cell function that occurs with aging [146,147]. Indeed, the potential use of biguanides, including metformin, as geroprotectors (substances that may extend lifespan and delay aging) was recognized as early as 1980 [148].

Metformin can activate AMPK, which directly inhibits several enzymes involved in ATP synthesis and decomposition, leading to a reduction in energy consumption. Additionally, metformin can modulate the peroxisome proliferator-activated receptor (PPAR) co-activator, potentially increasing mitochondrial biosynthesis. Furthermore, AMPK activation by metformin promotes autophagy, contributing to cellular health and maintenance [72]. Also, metformin has the potential to modulate the IGF-1 signaling pathway, resulting in decreased blood glucose levels, a potential slowdown of the aging process, and a potential extension of lifespan [149,150]. Metformin’s anti-aging effect may be achieved by inhibiting electron transport chain complex 1, leading to a decrease in reactive oxygen species (ROS) production. This, in turn, reduces the number of electron transfers and prevents electron leakage, resulting in lower ROS generation and reduced cumulative DNA damage [43,150].

The findings of a systematic review of 53 clinical studies investigating the effect of metformin on all-cause mortality suggested that metformin may have potential benefits in extending lifespan and improving overall health outcomes in individuals affected by age-related diseases, independent of its primary use for diabetes management [151].

Metformin altered various aging pathways, encompassing metabolic processes, collagen trimerization, extracellular matrix remodeling, adipose tissue metabolism, mitochondria, and MutS genes involved in DNA mismatch repair, as observed in the MILES trial involving 14 elderly participants with impaired glucose tolerance [152]. The trial’s results emphasize metformin’s capacity to impact diverse aging mechanisms. Besides this, the trial does not pinpoint the primary site of metformin action responsible for the observed changes in gene expression, leaving a critical aspect of its mechanism of action yet to be fully elucidated [152]. 

While preclinical and cell-based research has indicated positive outcomes, the findings from the majority of clinical studies present a more nuanced picture. The translation of promising results from laboratory settings to clinical applications has met with varying degrees of success, highlighting the complexity of applying interventions in real-world contexts. However, these findings await confirmation in additional research frameworks.

An overview of signaling pathways involved in neuroprotective mechanisms is shown in Figure 5.

## 4. The Winding Path from Neuroprotective Mechanisms to Proving Benefits in Neurodegenerative Diseases

The signaling pathways which can interfere with neurodegeneration are crucial to deciphering the complexities of neuroprotection and translate these discoveries into meaningful advances for individuals affected by neurodegenerative diseases. Long-term reactive oxygen species (ROS) produced in mitochondria during oxidative phosphorylation have been thought to be one of the main causes of aging-related DNA damage. Another process involves the constitutive activation of mTOR/S6 signaling, which is a sensor of nutrients and mitogens [153]. Aging inhibition affects the cellular pathways upstream of mTOR, including the PI3K, AKT, MAPK, and IGF-1/GH axis [154]. Inhibition of the IGF-1/GH axis reduces metabolic rate, cell proliferation, oxidative stress, decreases the accumulation of senescent cells, and counteracts inflammaging [155]. mTOR signaling has been associated with accelerated aging, and dysregulation of mTOR signaling has also been connected to the advancement of cancer, inflammatory and neurological illnesses, as well as T2DM [156]. Since mTOR is indirectly inhibited by AMPK activation, metformin, an AMPK activator, has been demonstrated to have gero-suppressive effects [157]. 

Cognitive dysfunction, which encompasses conditions like delirium, mild cognitive deficits, and dementia, signifies a significant deterioration from an individual’s previously achieved cognitive functioning level [158,159]. Some epidemiological studies have shown that metformin reduces the incidence of a variety of age-related illnesses as well as overall mortality. Importantly, this phenomenon was found in both diabetic and non-diabetic subjects [151,160]. A randomized, double-blind, placebo-controlled crossover study found that metformin had both metabolic and non-metabolic effects linked with aging in the elderly, indicating that metformin has an anti-aging impact [161]. Metformin has lately been proposed as an anti-aging medicine due to its widespread usage in clinical practice, well-known pharmacokinetics, and low toxicity.

Additionally, the impact of diabetes on neurodegeneration is notable, with chronic hyperglycemia and insulin resistance contributing to cognitive decline; however, metformin, beyond its previously discussed mechanisms, has shown promise in mitigating these effects and potentially exerting neuroprotective benefits in individuals with diabetes. An increasing number of epidemiological studies have identified cognitive impairment as a noteworthy comorbidity and complication of diabetes, thus elevating its status as a substantial public health concern [162,163,164]. A systematic review highlighted that individuals with diabetes face a 73% heightened risk of dementia and a 56% increased risk of AD [165]. Furthermore, the causes behind cognitive impairment in diabetes patients are likely multifaceted [166,167,168]. For instance, suboptimal glycemic control and the presence of microvascular complications such as neuropathy and retinopathy have been linked to cognitive dysfunction [169]. Diabetic macroangiopathy and microangiopathy have an interdependent relationship, where they exacerbate each other’s effects leading to inadequate blood flow in the brain and contributing to microvascular complications (cerebral atrophy, subcortical microinfarcts, lacunar infarction, and cerebral microhemorrhage) [170]. Neuronal injury, Alzheimer’s like pathologies, and the abnormal functioning of neurotransmitter receptors are closely connected to these vascular lesions [171,172,173]. Higher levels of blood glucose are a notable risk factor for inducing adverse alterations in the brain and causing impairment in brain function [174,175].The association between elevated blood glucose and progression of dementia is not restricted to DM alone but extends to conditions like prediabetes as well [33]. Maintaining normal blood glucose levels is important, but there is no proof endorsing aggressive glucose-lowering plans for individuals dealing with diabetes along with cognitive decline or memory loss [176,177,178].

Brain imaging studies conducted on individuals with T2DM have revealed a decrease in both overall and localized grey matter volume, including the volume of the hippocampus, in comparison to individuals without diabetes. This indicates potential structural alterations in the brain associated with T2DM [179,180].

This effect could be explained through various potential mechanisms. Research has demonstrated that metformin reduces insulin levels, ameliorates inflammation and oxidative stress and lowers the risk of thrombosis, consequently reducing the chances of developing metabolic syndrome [159,181,182,183]. Additionally, metformin enhances insulin sensitivity, which could potentially provide protection against cognitive dysfunction [159,184]. One of the most evident mechanisms by which metformin impacts the emergence of cognitive dysfunction in type 2 diabetes patients is by preventing hyperinsulinemia, a condition that might contribute to the formation of amyloid plaques in the brain and the onset of cognitive dysfunction [184].

Previous animal research suggested that metformin might improve cognitive impairment by reversing the adverse consequences of poor insulin signaling, which generates a cascade of negative events such as inflammation and tau hyper-phosphorylation [185,186]. Additionally, metformin’s positive effects may also result from its role in activating autophagy [11,187,188]. In a study led by Chen et al., male db/db mice at 12 weeks of age were given continuous intraperitoneal injections of 200 mg/kg/d of metformin, either alone or in combination with 10 mg/kg/d of chloroquine, for a span of eight weeks [145]. The administration of metformin to these mice alleviated cognitive decline, lessened the presence of p-Tau proteins, and reinstated impaired autophagy seen in diabetic mice [145].

Interestingly, these effects were reversed when chloroquine, an inhibitor of autophagic flux, was introduced, leading to a suppression of autophagy activity. In the context of HT22 cells cultured in a high-glucose environment, metformin exhibited a dose-dependent enhancement of autophagy. Importantly, it has been demonstrated that metformin’s augmentation of autophagy activity is contingent on the activation of AMPK [145]. In an experiment, metformin was found to reverse the learning and memory decline caused by scopolamine [189].This reversal does not seem to be connected to metformin’s influence on acetylcholine in the hippocampus. Instead, it could be attributed to its ability to enhance the levels of CREB and pAMPK that had been reduced by scopolamine. Additionally, metformin’s potential to counteract the impact of scopolamine on MDA, total antioxidant status, and superoxide dismutase (SOD) levels in the hippocampus might also contribute to its effectiveness [189]. 

In retrospective new-user research of 28,640 older veterans, it was discovered that age significantly affected the link between diabetes medication usage and dementia risk. Metformin was associated with an 11% lower risk of dementia than sulfonylureas in individuals younger than 75 years, while no difference was detected in those ≥75 years [178]. Similarly, a prospective observational study conducted in an Australian population revealed that the use of metformin by itself was associated with a decreased likelihood of developing cognitive dysfunction in individuals with type 2 diabetes [190].

Evidence suggests that metformin can potentially inhibit or slow down the onset of dementia in adults who have diabetes. In 2019, Shi and his team conducted a retrospective longitudinal cohort study concentrating on the impact of metformin in elderly adult US veterans diagnosed with T2DM and neurodegeneration [191]. The outcomes of this research revealed that undergoing metformin therapy for a duration of 2 to 4 years significantly lowers the risk of experiencing neurodegeneration in individuals with T2DM, in comparison to those who did not receive metformin treatment [191].

A meta-analysis that encompassed 10 studies (9 retrospective cohort studies and 1 prospective cohort study), including 254,679 patients, indicated that the use of metformin was connected to a notable decrease in the occurrence of cognitive impairment among individuals with type 2 diabetes [159]. Another meta-analysis released in 2020, which analyzed data from over 1 million patients, investigated the impact of different antidiabetic medications on the risk of dementia. This study indicated that specific antidiabetic agents like DPP-4 inhibitors and metformin are linked to a reduced risk of dementia in comparison to insulin [192].

Another meta-analysis suggests that metformin may enhance cognitive function in diabetic patients, potentially treating mild cognitive impairment and Alzheimer’s disease, although its use for dementia prevention in older non-diabetic adults is not currently recommended, according to the study [193].

Conversely, studies conducted by Koo and his team demonstrated contrary results. These studies indicated that metformin treatment was not effective and, in fact, might have exacerbated the cognitive condition in elderly Korean patients [194]. A recent meta-analysis indicates that metformin does not exhibit a significant impact on enhancing cognitive function or providing protection against any forms of dementia, including both vascular dementia and AD [35]. This conclusion extends to cognitive impairment as well [35].

Similarly, no strong evidence was identified in a Cochrane meta-analysis of seven studies to suggest that any particular type 2 diabetes therapy or treatment plan can prevent or postpone cognitive impairment [176].

Furthermore, the occurrence hypoglycemia events is linked to a decrease in cognitive function and a higher likelihood of developing dementia [195,196]. Regarding the ways in which metformin controls hypoglycemia, existing research focuses on the activation of AMPK and the enhancement of gut glucagon-like peptide 1 (GLP-1) levels [197,198].

While the prevailing data regarding the use of metformin in cases of dementia, with or without T2DM, tends to be positive, it is crucial to recognize that the impact of metformin could be contingent on intricate underlying pathological mechanisms [16].

### 4.1. Alzheimer’s Disease

The underlying causes of AD are still not fully understood, which presents a difficulty in developing new and effective treatment approaches [199]. AD is characterized by amyloid plagues and neurofibrillary tangles in the brain with a broad range of dementia symptoms, including memory loss, language difficulties (aphasia), difficulties in carrying out 160 motor tasks (apraxia), challenges in recognizing familiar objects or people (agnosia), impaired visual–spatial skills, problems with executive function, and noticeable changes in personality and behavior [200,201,202]. The need for research to find therapies has grown due to the rise in AD incidence globally. Behavioral and psychological symptoms, including aggression, significantly contribute to the social and caregivers’ burden and represent a major challenge. A systematic review of six clinical trials involving 422 patients evaluated the efficacy and safety of psychoactive cannabinoids in treating psychiatric manifestations, including agitation and aggression from AD, but no conclusive conclusion was reached [203]. Aggressive behavior frequently imposes patient hospitalization and atypical antipsychotic drugs are needed to manage the symptoms [204].

Increasing evidence strongly indicates that both diabetes and AD share common characteristics of insulin resistance, hyperinsulinemia, oxidative stress, inflammation, and mitochondrial dysfunction [205,206]. T2DM and late-onset Alzheimer’s disease (LOAD) have emerged as global epidemics, and current projections suggest that their prevalence will increase in the upcoming years. LOAD, which has recently been characterized as a metabolic disease because obesity, T2DM, and their associated comorbidities have been linked to its progression, is associated with ineffective use of glucose in the nervous system as well as resistance to insulin and a persistent state of mild inflammation within the brain [207,208,209,210,211,212]. LOAD has also been commonly referred to as “type 3 diabetes” because of the insulin resistance that exists in the brain [207,211,213].

Several studies provide confirmation that metformin exerts a positive influence on human neural stem cells. In the study by Wang et al., it was found that metformin inhibits the formation of the caveolin1/AMPKa complex. This inhibition leads to the activation of AMPKa, which ultimately results in the suppression of inflammation and endoplasmic reticulum stress induced by high glucose levels [138]. Research involving human neural stem cells (hNSCs) exposed to Aβ (amyloid-beta) has revealed that the use of an AMPK agonist like metformin resulted in decreased cell viability and an increase in the activity of caspase 3/9, which serves as an indicator of caspase cascade activation [214]. Cells subjected to Aβ demonstrated that Aβ also plays a role in triggering apoptosis by facilitating the distribution of cytochrome c, a crucial factor that induces apoptosis by activating caspase 9. This cytochrome c is released from the mitochondria into the cytosol. Remarkably, the application of metformin effectively prevents this release of cytochrome c [214]. In research conducted by Chung, hNSCs that were exposed to AGEs (advanced glycation end products) experienced a notable decrease in cell viability. This decrease was correlated with reduced expressions of AMPK and genes/proteins associated with mitochondria (PGC1α, NRF-1, and Tfam) [215]. Additionally, there was an elevation in the activation of caspase 3 and 9 activities. Importantly, the administration of metformin effectively hindered the release of cytochrome c from the mitochondria into the cytosol in the hNSCs that were influenced by AGEs [215].

Numerous studies have provided evidence supporting the beneficial effects of metformin in enhancing cognitive, memory, and learning functions in animal models of AD [135,216]. The evidence indicates that metformin has the capacity to prevent or alter the aggregation process of Aβ, improve mitochondrial function, promote neurogenesis, and exhibit antioxidant and anti-inflammatory effects in various animal models of AD [32,217,218]. Metformin was found to improve plaque-associated tau pathology and reduce the accumulation of Aβ in the hippocampus and cortex of the brains of APP/PS1 mice [135,219]. Lu et al. reported that in APP/PS1 mice, the administration of metformin resulted in the attenuation of the increase in superoxide dismutase (SOD) activity and malondialdehyde (MDA) level [134]. Furthermore, it has been confirmed that metformin can regulate the release of glutamate on the presynaptic membrane, and that this effect occurs in an AMPK-independent manner [220]. In an in vivo study using SAMP8 mice, which is a model for sporadic AD, metformin was found to reduce the levels of APPc99 and pTau404 expression. These changes were associated with improvements in mitochondrial function, as observed in previous research studies [186,221,222].

Based on research conducted on transgenic mice models including APPswe, PS1dE9, and PDAPP (J9), as well as AD models triggered by chemical induction like sporadic AD models induced by STZ, metformin has been shown to prevent damage to the hippocampus and the decline in spatial memory [211]. Additionally, metformin reduces inflammation and modulates the AMPK/mTOR/S6K/Bace1 pathway [211]. Moreover, metformin has been observed to enhance the sensitivity of insulin receptors and support the survival of neurons in these contexts [211]. In another study, it was found that metformin preserves the plasticity of hippocampal synapses and restores the cleavage of acetylcholine in the cortex and hippocampus. This action leads to a decrease in the activity of acetylcholinesterase in in vitro experiments [223]. Moreover, metformin was shown to reverse spatial learning and memory impairments in mice with AD induced by STZ (streptozotocin), and it also normalized glucose transportation in their brain [223].

As stated before, in an experimental study utilizing a model of AD involving double transgenic mice (APP/PS1), the administration of metformin (at a dose of 200 mg/kg via intraperitoneal injection) was able to reverse spatial memory impairments. This treatment also served to prevent neuronal cell death and promote the growth of new neurons in the adult hippocampus, as indicated by increased neurogenesis [135].

Sexual dimorphism can indeed impact the effects of metformin, possibly making it more difficult to evaluate animal data. Male rodents are frequently prioritized in research, occasionally resulting in the oversight or exclusion of the sex of the animals used. In a previously mentioned metformin study, male mice exhibited impaired cognitive function following treatment, whereas female mice showed improvements after treatment. This underlines the significance of considering sex as a crucial variable when assessing the outcomes of metformin treatment in animal models [224].

In a recent meta-analysis involving 285,966 participants, metformin was found to have a neutral effect on the risk of AD, meaning it did not significantly increase or decrease the risk of AD. However, the meta-analysis revealed that metformin was associated with an increased risk of PD [225]. Another meta-analysis conducted by Campbell et al., the utilization of metformin among individuals with diabetes was associated with a decreased risk of developing dementia and AD [168]. Within the collection of 14 studies that were encompassed in this meta-analysis, three investigations identified positive impacts of metformin on cognitive function [168]. Additionally, six studies presented evidence indicating that the risk of dementia was notably lowered among diabetes patients who incorporated metformin into their treatment regimen, its protective effects have not been confirmed in non-diabetic patients [168].

In their study among elderly US veterans with T2DM, Shi et al. found that using metformin for less than one year was associated with an elevated risk of AD when compared to individuals not taking metformin. However, taking metformin for more than four years was linked to a protective effect against AD in patients with T2DM [191]. Sluggett et al. observed comparable findings. Taking metformin for a duration of 1 to 3 years was associated with an increased risk of AD. On the other hand, for aged patients with T2DM, long-term use (≥10 years) of metformin and using high doses (an average of 2g daily) showed a protective effect against the development of AD [226].

Columbia University in New York City conducted a preliminary study between 2008 and 2012 involving 80 overweight, non-diabetic individuals diagnosed with amnestic moderate cognitive disorder [227]. The study assigned them either 2000 mg of metformin split into two doses or a placebo over the course of one year. The primary finding of the study indicated that metformin led to an enhancement in recall efficacy, particularly in the selective reminding test, but no distinctions were observed in the Alzheimer’s disease assessment scale-cognitive subscale (ADAS-Cog) scores, glucose uptake, or plasma levels of Aβ42 between the two groups [227].

A randomized, double-blinded, placebo-controlled study was conducted between 2013 and 2015 to explore the impact of metformin on various aspects of AD, including biochemical, neurophysiological, and cognitive indicators. Koenig and colleagues demonstrated that metformin was able to cross the blood–brain barrier, as indicated by measurable levels in the cerebrospinal fluid [228]. However, over the 8-week exposure period, it did not show a noticeable effect on CSF AD biomarkers. Intriguingly, the study observed enhancements in executive functioning during metformin treatment but not during the placebo phase. Trends also hinted at improvements in learning, memory, and attentional capabilities during metformin treatment [228].

A multicenter phase 2/3 prevention trial known as Metformin in Alzheimer’s Dementia Prevention (MAP) started in March 2021. This study will involve the enrolment of 370 individuals across nine academic medical centers in the United States. Eligible participants must be aged between 55 and 90, be overweight or obese, without diabetes, and exhibit either early or late mild cognitive impairment (MCI). Individuals with normal weight, who are less likely to exhibit a metabolic response to metformin, will be excluded from the study. Over a span of two years, participants will be administered 2000 mg per day of extended-release metformin or a placebo. The main measure of interest is the change in scores on the free and cued selective reminding test. Additionally, secondary outcomes encompass alterations in the Alzheimer’s disease cooperative study preclinical Alzheimer’s cognitive composite (PACC-ADCS), changes in hippocampal volume, and variations in white matter hyperintensity volume. The trial is projected to continue until 2026 (https://classic.clinicaltrials.gov/ct2/show/NCT04098666) (accessed on 10 August 2023).

Contradictory findings were found by several investigations. Research indicates that metformin increases the expression of APP and presenilin-1 genes, leading to increased intra- and extracellular Aβ levels through free radical production, mitochondrial malfunction, and cell death [229]. This suggests metformin may be involved in the oxidative stress-induced synthesis of Aβ peptides [229].

Various studies have revealed negative consequences of metformin on the risk of AD in different animal models, which contradicts the results presented above. For example, a recent study by Kuhla et al. revealed that metformin enhances lipogenesis, leading to fat accumulation, triggers neuroinflammation, and reduces neuronal integrity. These cumulative effects aggravated tauopathy and exacerbated manifestations related to AD in ApoE^−/−^ mice [230]. A study by Peng et al. has shown conflicting results. It stated that metformin injection decreased blood glucose in adult mice and reversed the hyperglycemic effect of isoflurane during anesthesia [231]; however, glycemic control did not improve hippocampus-dependent fear memory, and gavage treatment with 25% glucose solution elevated blood glucose even under metformin treatment. The mechanism behind acute hyperglycemia’s cognitive decline remains unclear [231].

According to a recent review, the age of animal models, treatment durations, dosages, and experimental methods all affect the variability observed in studies. Clinical studies are required to understand molecular pathways [221]. In AD patients, cognitive impairment persists even after β-amyloid peptide levels are lowered. Although the Aβ cascade theory has gained popularity, it remains unclear how it relates to AD [221]. Clinical studies in phase III have not succeeded, suggesting that lowering beta amyloid does not markedly enhance cognitive abilities [221].

Imfeld et al. discovered that prolonged metformin use elevated the likelihood of AD onset when compared to control groups [232]. Notably, in this study, the extended usage of other anti-diabetic medications, including sulfonylureas, thiazolidinediones, and insulin, did not yield any discernible impact on the risk of AD development [232]. Moreover, Moore and associates demonstrated that metformin led to a deterioration in cognitive function [233]. The study by Wu et al. discovered that metformin had no notable effect on T2DM patients’ cognitive impairment [234].

Furthermore, both preclinical research and clinical studies have indicated a connection between T2DM and not only the onset but also the advancement of LOAD. A meta-analysis of 20 prospective observational studies concluded that the occurrence of LOAD in individuals with T2DM was 56% higher compared to those without diabetes [34]. On the contrary, there are counter arguments asserting that diabetes does not necessarily predispose individuals to AD [235]. Several clinical studies have not been able to establish a strong link between AD and diabetes, except in the case of carriers of the apolipoprotein E (ApoE) ε4 allele who are twice as likely to develop AD as those without diabetes [236,237,238].

While a substantial volume of evidence supports the beneficial impact of metformin on reducing the risk of AD, there are studies presenting contradictory results. Given the significance of factors such as dosage and duration of exposure, conducting more retrospective and prospective studies can offer deeper understanding regarding the most effective and appropriate utilization of metformin for addressing this purpose.

### 4.2. Parkinson Disease

About 1–2% of people 65 years of age and older have PD, the second most common neurological condition, and currently lack disease-modifying treatments [239,240]. PD is characterized by motor symptoms like bradykinesia, rigidity, and resting tremor, alongside non-motor symptoms such as dementia and depression. The main pathological features involve the degeneration of dopaminergic neurons in the substantia nigra pars compacta and the presence of Lewy bodies (LB), which are protein aggregates mainly composed of α. These alterations affect the transmission of neurotransmitters in the nigrostriatal pathway [241].

Despite considerable progress in PD research in recent years, the specific causes of PD are still not fully understood. However, an increasing amount of evidence suggests that dysregulated autophagy, protein aggregation, oxidative damage, neuroinflammation, and the aging process all contribute significantly to the development of PD [242,243]. Increasing evidence suggests that patients with T2DM have a heightened risk of PD and exhibit overlapping dysfunctional pathways, implying a potential common underlying pathological mechanism between the two conditions [244]. Additionally, the presence of further evidence suggests that 50% of patients with PD exhibit signs of amyloid-β peptide plaques and hyperphosphorylated tau-containing neurofibrillary tangles, which are typically observed in the brains of individuals with AD [245,246]. This evidence raises the possibility that metformin may have potential beneficial effects in addressing these overlapping pathologies in PD patients.

Metformin may treat PD symptoms through the following mechanisms: (1) metformin-induced protein kinase B activation may activate nuclear factor Nrf2/heme-oxygenase-1 pathway to ameliorate mitochondrial dysfunction and energy deficit [247,248], (2) metformin may reduce inflammation by suppressing microglial activation and decreasing proinflammatory cytokines (TNF- and IL-1) [249,250,251], (3) metformin may prevent dopaminergic neuron loss [250,251], (4) Metformin may inhibit α-synuclein phosphorylation [252].

A multitude of studies have presented compelling evidence affirming the advantageous outcomes of metformin in augmenting cognitive, memory, and learning functions within animal models of PD. Lu et al. discovered that metformin exhibited a protective effect on neurodegeneration by inhibiting overactivated microglia-induced neuroinflammation in the substantia nigra compacta of mice with 1-methyl-4-phenyl-1,2,3,6-tetrahydropyridine (MPTP) plus probenecid-induced PD. Additionally, metformin activated AMPK in SH-SY5Y cells, leading to microtubule-associated protein 1 light chain 3-II-mediated autophagy and the removal of mitochondrial reactive oxygen species [253]. In the mouse model of PD using 6-hydroxydopamine (6-OHDA) lesions, metformin was found to inhibit the development of dyskinesia and modulate Akt and glycogen synthase kinase 3 (GSK3) signaling, as well as astrocyte activation [249,254]. In the LPS-induced rat model of PD, metformin was observed to primarily inhibit the activation of microglia and reduce the expression of inflammatory cytokines [255]. In the 6-hydroxydopamine (6-OHDA) mouse model of PD, the co-treatment of metformin with L-dopa was found to improve L-dopa-induced dyskinesia.

In one study published in 2023, the comparative neuroprotective capabilities of metformin and trehalose were assessed for the first time in a paraquat (PQ)-induced PD model with epidemiological relevance [256]. The study revealed that both metformin and trehalose, which act as autophagy inducers, exhibited antioxidant effects, enhanced mitochondrial activity, and reduced the death of dopaminergic cells induced by PQ in vitro [256]. Furthermore, in vivo, both autophagy inducers mitigated cognitive and motor function decline, preserved dopaminergic neurons and oligodendrocytes, reduced astrocytosis and microgliosis, and prevented cell death triggered by PQ [256]. Notably, the induction of autophagy was achieved through pAMPK by both metformin and trehalose, leading to a decrease in α-synuclein accumulation, a protein associated with PD pathology [256].

A recent randomized, double-blind, placebo-controlled clinical trial unveiled that exenatide, an agonist of GLP-1, could potentially yield beneficial outcomes in terms of motor symptoms for individuals with PD [257]. This evidence suggests that the increase in GLP-1 by metformin could be another potential molecular mechanism contributing to its neuroprotective role in PD [16,254]. Notably, metformin did not interfere with the pharmacotherapeutic effects of L-dopa, suggesting that the combination of both treatments might be beneficial in managing dyskinesia in PD without compromising the efficacy of L-dopa [254].

Conversely, a different experiment employing an inflammatory model of PD revealed that metformin not only failed to prevent the death of dopaminergic neurons triggered by a lone intranigral injection of LPS, but it also exacerbated the harm [255]. For instance, in the context of neurodegeneration, metformin was found to inhibit the brain’s inflammatory response induced by MPTP, affecting the microglia polarization state by targeting iNOS, IL-1β, and TNF-α [258]. However, it should be noted that MPTP and metformin might have an additive effect on inhibiting complex I of the electron transport chain in PD patients, leading to reduced ATP levels [258]. Moreover, extended use of metformin has been linked to hyperhomocysteinemia, as well as deficiencies in vitamin B12 and folate [259]. A recent systematic review and meta-analysis indicated that metformin has a neutral effect on homocysteine levels in T2DM patients when they are supplemented with B12 and folate [260].

Three studies have reported an elevated risk of PD among patients who were taking metformin [261,262,263]. According to a systematic review published in 2020, which analyzed 23 comparisons from 19 studies involving over 250,000 subjects, no beneficial effects were found associated with the use of metformin for PD. On the contrary, the review indicated that metformin use might even worsen the risk of PD [225]. In a comprehensive analysis conducted by Liu et al., which included seven case–control studies involving 26,654 PD patients and nine cohort studies with 3,819,006 DM patients, it was revealed that individuals with diabetes have a 15% higher risk of developing PD [264]. On the other hand, a meta-analysis published in 2023, encompassing fifteen studies and involving a total of 2,910,405 diabetic patients, found that antidiabetic medications including metformin, sulfonylureas, glucagon-like peptide-1 agonists, and glitazones did not display any significant association with alterations in the risk of PD [265].

Additionally, a clinical trial proposed that the reduction in serum vitamin B12 associated with metformin could be attributed to the increased transportation and utilization of vitamin B12 by cells, which is stimulated by metformin [266] and the correlation between low-normal levels of serum vitamin B12 and PD has been established [267,268,269,270].

In their meta-analysis, Qin et al. stated that the contradictory results between studies can be related to inter-study variability in demographics, drug doses and duration, medication classes, follow-up lengths, and adjusted covariates [271]. Based on the varying perspectives, the prevailing view suggests that metformin likely exhibits a protective effect rather than a detrimental one in PD neuropathology. Nevertheless, it is essential to conduct clinical trials and studies specifically focusing on non-diabetic subjects to further investigate the impact of metformin on PD risk.

### 4.3. Huntington

Huntington’s disease (HD) is a neurodegenerative condition characterized by diminished motor ability, chorea (involuntary motions), and a progressive decline in cognitive function. HD patients have an abnormal CAG expansion within the first exon of the huntingtin gene, HTT [272]. In the presence of toxic mutant huntingtin (mHtt), neurons initiate pathways of protein clearance, such as autophagy [273,274] or the proteasome. AMPK, a key regulator of eukaryotic cell energy balance, is among the molecules that can trigger autophagy and plays a significant role in this process [53,275,276].

It has been shown that metformin’s activation of AMPK reduces cell death in striatal cell precursors under stress brought on by mHtt and improves neuronal dysfunction in worms expressing polyglutamine [277]. Metformin also activates the phosphatase PP2A [30]. The inhibition of mTORC1 by metformin-activated AMPK and PP2A could lead to several beneficial effects in HD [278]. One potential mechanism is the enhancement of basal autophagy, which could contribute to the positive outcomes observed [278]. Sanchis et al. found that metformin, administered during early HD stages in zQ175 mice, alleviates neuropsychiatric and motor symptoms, potentially mitigating neuronal toxicity from mHtt aggregation in the striatum and cortex [279]. Remarkably, metformin has been reported to reduce the accumulation of aberrant Huntingtin protein and completely restore the heightened synchronicity and hyperactivity of neurons, resulting in the reversal of behavioral abnormalities [30]. Arnoux et al. conducted a study to investigate the importance of reducing mHTT levels in HD pathology. Metformin reduces aberrant Huntingtin protein load and completely restores both early network activity patterns and behavioral aberrations in cortical neuronal microcircuits of premanifest Hdh150 mice, serving as a model for a ‘very far from disease onset’ stage in the human illness [30].

A post hoc statistical analysis of participants from the Enroll-HD database (https://www.enrollhd.org, accessed on 30 August 2023) compared the cognitive performance of HD motor manifest patients taking metformin to similar patients not taking metformin [280,281]. The results of this analysis indicated that metformin intake was associated with improved cognitive performance as evaluated through various cognitive tests.

The multifaceted effects of metformin and the absence of a specific target directly related to the etiology of HD may naturally make metformin a less enticing therapeutic option for HD.

### 4.4. Epilepsy

Around 70 million people are affected by epilepsy, which has become a severe worldwide health problem. Comorbidities, or related diseases that often coexist with epilepsy, make the situation worse [282]. It is well recognized that individuals with epilepsy (PWE) have a greater risk of developing cognitive impairment, depression, anxiety, schizophrenia, autism [283,284], as well as AD [285,286,287]. Antiseizure drugs (ASDs) are the primary treatment for epilepsy. Despite the variety of available ASDs, around a third of patients are unable to manage their seizures or develop resistance to the drugs [288,289]. Therefore, there is a pressing requirement for the advancement of novel antiepileptic treatment approaches to enhance disease management and mitigate its adverse effects.

While the exact mechanism underlying the anti-epileptic properties of metformin remains largely unclarified, there is an increasing comprehension that its anti-seizure potential predominantly stems from its ability to mitigate oxidative damage in the brain, activate AMPK, suppress the mTOR pathway, reduce α-synuclein expression, inhibit apoptosis, and downregulate levels of BDNF and TrkB [290,291,292,293,294,295,296,297].

As certain types of epilepsy are associated with an increased activity of the mTOR pathway, various therapeutic approaches have been developed to suppress this pathway [298,299]. The activation of AMPK also prompts immune cells to transition from a pro-inflammatory state to an anti-inflammatory one [300,301]. On the contrary, the mTOR pathway is stimulated in situations of high-energy availability, engaging in the activation of processes that build molecules and restraining those that break them down. Notably, there exists an intriguing interplay between the AMPK and mTOR pathways, whereby the activation of AMPK leads to the suppression of the mTOR system. Interestingly, metformin’s activation of AMPK inhibits mTOR signaling, and this has demonstrated enhanced seizure management in scenarios characterized by mTOR hyperactivity [58,290,302].

Activation of AMPK also confers advantageous outcomes in alternative epilepsy models. In specific instances, metformin demonstrates a positive impact on seizures and epilepsy in mice subjected to induced epilepsy through pentylenetetrazol (PTZ), a pro-convulsive agent. This effect is attributed to metformin’s antioxidative and anti-apoptotic properties, as well as its augmentation of heat shock protein 70 [290,291,294].

The disruption of the blood–brain barrier (BBB) is hypothesized to lead to heightened permeability to pro-inflammatory cytokines, ultimately contributing to the occurrence of seizures [303,304,305]. BBB dysfunction has been linked to conditions like AD [306], post-traumatic epilepsy [307], psychiatric disorders, and cognitive impairment in type 2 diabetes [308]. Research has indicated that metformin could mitigate the risk of seizure development in individuals with diabetes by mitigating BBB disruption, primarily through its anti-inflammatory actions as previously discussed [309].

Moreover, numerous antioxidants have showcased their potential as antiepileptic agents in experimental studies, primarily by offering protection against oxidative damage [310,311]. Metformin has also exhibited its ability to mitigate the oxidative damage induced by epilepsy in experimental studies which point to the potential involvement of oxidative stress in the pathophysiology of PTZ-induced seizures, suggesting that the antiepileptic impact of metformin is linked to its antioxidative properties [291,294].

Despite the considerable evidence from pre-clinical studies showcasing the effectiveness of metformin as a promising antiepileptic agent in models involving kainic acid, PTZ, and pilocarpine, extensive clinical trials on a broader scale are still required to ascertain its viability for human use. As of now, there has been one randomized controlled trial carried out involving human participants with tuberous sclerosis, with a notably limited sample size (*n* = 51). This study demonstrated the advantageous impact of metformin in a specific subgroup of individuals with epilepsy, specifically those afflicted with tuberous sclerosis [312]. An additional retrospective study examined the effectiveness of metformin in individuals with genetically confirmed Lafora disease, revealing that it lowered the frequency of seizures and led to an enhancement in the overall clinical condition of the patients [54,313]. Nonetheless, additional research is necessary to facilitate the clinical application of metformin for the treatment of epilepsy.

### 4.5. Fragile X Syndrome

Fragile X syndrome (FXS) stands as the most prevalent type of hereditary cognitive impairment, impacting approximately 1 in 5000 males and 1 in 6000 females on a global scale [314]. It is attributed to a mutation in the FMR1 gene located on the X chromosome. This mutation results in the absence or reduced production of a protein known as fragile X mental retardation protein (FMRP), which plays a crucial role in normal brain development and functioning [315].

In the context of the Drosophila model of Fragile X Syndrome (FXS), characterized by dfmr1 mutations, researchers recently identified increased insulin signaling in the brain. Through the utilization of Drosophila as a FXS model, it was demonstrated that targeting the expression of dfmr1 specifically in the brain’s insulin-producing cells (IPCs) effectively reversed abnormal circadian behavior and alleviated memory impairments in the Fragile X mutant flies [316]. Furthermore, a short-term administration of metformin resulted in the improvement of deficits in olfactory learning as well as long-term memory [316].

Moreover, one research investigation demonstrated that administering metformin to Fmr1KO mice for a period of 10 days at a dose of 200 mg/kg/day corrected behaviors like excessive grooming and impaired social interactions, lowered the occurrence of audiogenic seizures, and partially reversed changes in testicular weight [317]. However, metformin did not impact hyperactivity. Additionally, the study found that metformin intervention rescued abnormal long-term depression and restored spine morphology. There were also improvements in ERK signaling, but there was not any observed effect on mTOR signaling [317].

In a case series of seven individuals with FXS who were treated with metformin, there were consistent positive changes noted. Particularly, enhancements were noted in irritability, social responsiveness, hyperactivity, and social avoidance among these individuals with FXS [318].

## 5. Concluding Remarks and Future Prospects

Metformin, discovered 60 years ago, is an insulin-sensitizing drug that can lower peripheral insulin resistance and is used as a first-line therapy in the treatment of type 2 diabetes, with proven benefits and an advantageous safety profile. Despite extensive research, the precise molecular mechanisms of its therapeutic effects are not fully elucidated. Activation of the LKB1(Liver kinase B1)–AMPK signaling pathway, considered for a long time the main mechanism explaining the therapeutic effects of metformin on the glucose metabolism, plays an important role in chronic metformin administration. The pleiotropic impact of metformin may involve other stress kinases that are acutely activated. Metformin interacts with mitochondrial and respiratory chain complexes I and IV, which slow oxidative phosphorylation and decrease gluconeogenesis. This process helps reduce oxidative stress on neurons by minimizing the use of NADH. The precise role of different molecular targets such as the respiratory chain complexes, mitochondrial targets or the lysosomes in metformin effects remains to be established by further research. The roles of gut and microbiota, as an important extrahepatic target of metformin’s effect, as well as its immunomodulatory properties, are also to be further explored, in close relation with pathogenic mechanisms in neurodegenerative disorders.

Furthermore, metformin promotes neurogenesis, angiogenesis, and synaptic plasticity by inducing autophagy. Metformin protects the brain by controlling cognitive impairment, neurogenesis, mitochondrial malfunction, oxidative stress, aging, and autophagy.

Several investigations have concluded that metformin has a positive effect on the prevention of neurodegenerative disorders. Although most of the data on this subject is observational, the number of randomized studies is growing. Previous studies on animals revealed that metformin might enhance cognitive impairment by correcting the detrimental effects of inadequate insulin signaling, which leads to a chain reaction of unfavorable events including inflammation and tau hyperphosphorylation. Regarding AD, animal-based studies draw the conclusion that metformin has a positive effect, while observational studies state that short-term use increases the risk of AD and long-term use has a protective effect.

The beneficial effects of metformin on enhancing cognitive, memory, and learning abilities have been well supported by research in animal models of PD and a few clinical trials. On the other hand, metformin was associated with an increased risk of PD in some studies, possible due to deficiencies in vitamin B12 and folate related to its administration.

Despite the challenges posed by its complex effects and the lack of a specific etiological target in HD and FXS, metformin, being an approved drug with generally well-tolerated side effects, still holds the potential to be a significant disease-modifying option for these hereditary neurodegenerative disorders. Significant research questions are still open, nevertheless. Since patients with diabetes make up the majority of the outcomes reported in the literature, it is crucial to fully understand the possible benefits of metformin for the general non-diabetic population. On the other hand, understanding the molecular targets and mechanisms of its effects could help design the clinical trials to explore and, hopefully, prove its therapeutic effect in neurodegenerative conditions.

## Figures and Tables

**Figure 1 pharmaceuticals-16-01714-f001:**
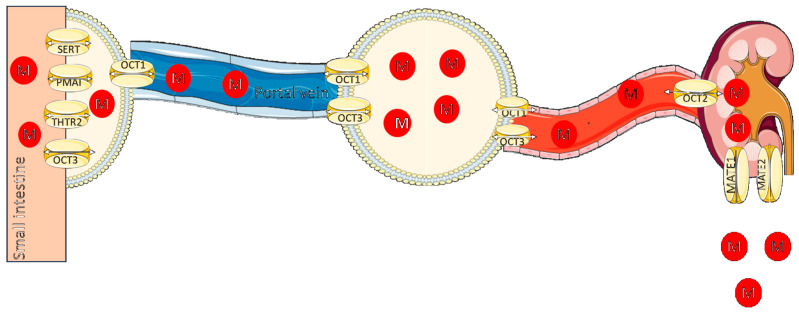
Overview of transporters relevant to metformin pharmacokinetics: the MATE multidrug and toxin extrusion transporter, OCT organic cation transporter, PMAT plasma membrane monoamine transporter, SERT serotonin transporter, THTR2 thiamine transporter. Parts of the figure were drawn by using pictures from Servier Medical Art. Servier Medical Art by Servier is licensed under a Creative Commons Attribution 3.0 Unported License (https://creativecommons.org/licenses/by/3.0/).

**Figure 2 pharmaceuticals-16-01714-f002:**
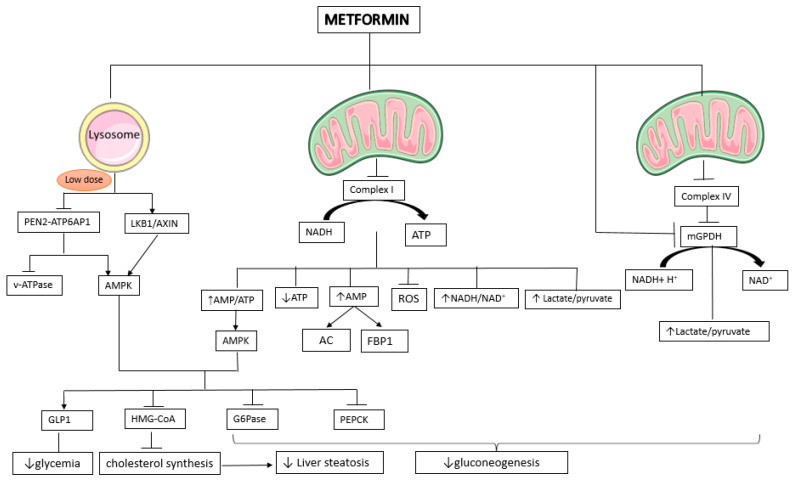
Proposed mechanism of action for metformin. Left: A low dose of metformin interferes with the vacuolar H+-ATPase in the lysosomal proton pump (v-ATPase). The interaction between metformin and PEN2 (presenilin enhancer protein 2) forms a complex involving ATP6AP1. The AMPK-dependent model of metformin-induced inhibition of gluconeogenesis is based on the activation of the LKB1–AMPK signaling pathway. Middle: Metformin causes a slight inhibition of mitochondrial respiratory chain complex I in the liver, which results in a moderate reduction in ATP production and an increase in AMP (inhibiting Fructose-1,6-Bisphosphatase Deficiency FBP1 and adenylate cyclase AC) levels in the cells. AMP-activated protein kinase (AMPK) is activated by the metformin-induced rise in the AMP-to-ATP ratio and increases GLP1 and inactivates/inhibits G6Pase(glucose-6-phosphatase), PEPCK (phosphoenolpyruvate carboxykinase), and HMG-CoA (3-hydroxy-3-methylglutaryl coenzyme A). The inhibition of complex I by metformin is also accompanied by an increased ratio of lactate/pyruvate, and reduced glucose synthesis. Right: Mitochondrial glycerol-3-phosphate dehydrogenase (mGPDH)-dependent. Because metformin directly inhibits mGPDH, there is a decrease in lactate gluconeogenesis, a reduction in the activity of the glycerol–phosphate shuttle (which moves NADH from the cytosol to the mitochondria), and a rise in the cytosolic redox state (NADH:NAD+). “↑” increase, “↓” decrease. Parts of the figure were drawn by using pictures from Servier Medical Art. Servier Medical Art by Servier is licensed under a Creative Commons Attribution 3.0 Unported License (https://creativecommons.org/licenses/by/3.0/).

**Figure 3 pharmaceuticals-16-01714-f003:**
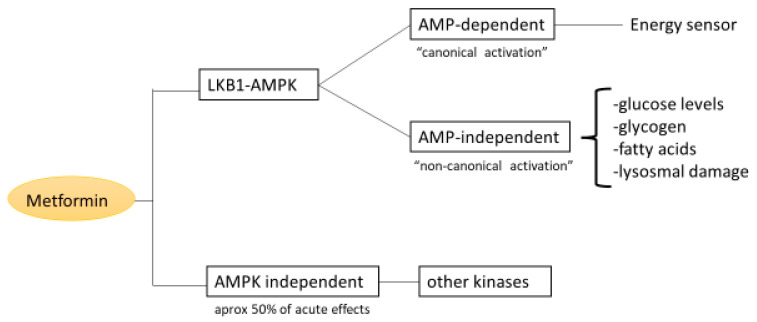
AMPK signaling pathway. AMPK AMP-activated protein kinase, LKB1 Liver kinase B1.

**Figure 4 pharmaceuticals-16-01714-f004:**
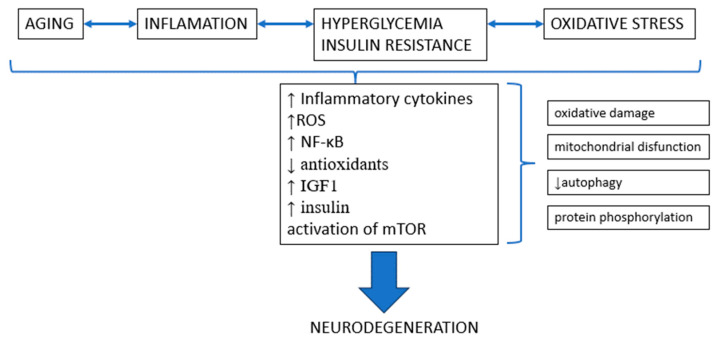
Mechanisms involved in neurodegeneration. “↑” increase, “↓” decrease.

**Figure 5 pharmaceuticals-16-01714-f005:**
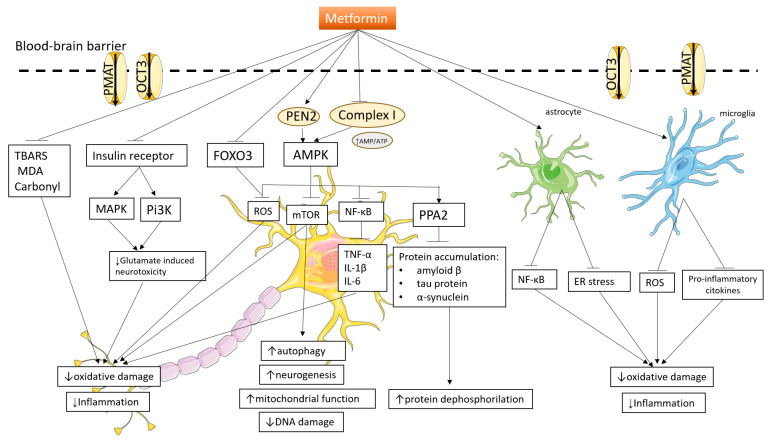
Signaling pathways involved in neuroprotective mechanisms. Metformin’s potential as a neuroprotective agent. Metformin is known to traverse the BBB through the actions of the PMAT and OCT3 transporters. Metformin modulates inflammatory status and inhibits oxidative damage markers like TBARS, MDA, carbonyl groups, and FOXO3, while concurrently stimulating AMPK. Furthermore, the activation of AMPK by metformin determines the inhibition of mTOR, resulting in increased autophagy, neurogenesis, enhanced mitochondrial function, and decreased DNA damage. This AMPK activation also plays a crucial role in reducing protein phosphorylation by inhibiting PPA2. Notably, metformin’s neuroprotective influence extends beyond neurons, encompassing astrocytes and microglia, effectively curtailing oxidative damage and inflammation in the neural web. Legend: BBB: blood–brain barrier; PMAT: plasma membrane monoamine transporter; OCT3: organic cation transporter 3; TBARS: Thiobarbituric acid reactive substances; MDA: Malondialdehyde; FOXO3: forkhead box O3; AMPK: AMP-activated protein kinase; mTOR: mechanistic target of Rapamycin; PPA2: Protein Phosphatase 2A; MAPK: mitogen-activated protein kinase, TNF- α; tumor necrosis factor α; NF-kB: nuclear factor-kappa B; ER: endoplasmic reticulum; PI3K: Phosphoinositide 3-kinase, -1β interleukin 1 beta, IL-6 interleukin 6; “↑” increase, “↓” decrease. Parts of the figure were drawn by using pictures from Servier Medical Art. Servier Medical Art by Servier is licensed under a Creative Commons Attribution 3.0 Unported License (https://creativecommons.org/licenses/by/3.0/).

## Data Availability

Data sharing is not applicable.

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
