# Peer review of "Metformin: The Winding Path from Understanding Its Molecular Mechanisms to Proving Therapeutic Benefits in Neurodegenerative Disorders"

_pharmaceuticals, 2023, doi:10.3390/ph16121714_

Round 1
Reviewer 1 Report
Comments and Suggestions for Authors
"Narrowing the gap between in vitro and in vivo studies regarding metformin's neuroprotective effects" is a review about neurodegenerative diseases and the potential of metformin as neuroprotective agent.
The review is conducted according to the state of art reaching very recent date (last month August 2023).
Taking account of the broadness of the analysis and the rich bibliography on the topic the manuscript is worth publishing.
Author Response
Response to Reviewer 1:
Summary:
First, we thank to editors and all the reviewers for their effort and the time spent to improve our work and for the insightful comments on our article. Below, we respond to each comment in an itemized fashion, hoping that the related changes to the manuscript are satisfactory, so that the paper can be accepted in its currently revised form.
The modifications made to the manuscript have been marked in the manuscript.
Response to Reviewer 1:
Thank you for your positive comments.

Reviewer 2 Report
Comments and Suggestions for Authors
General Comments:
The manuscript provides a comprehensive review of the potential neuroprotective effects of metformin, a widely prescribed medication for type 2 diabetes. The authors have done a commendable job in collating and analyzing the available evidence, both from experimental and clinical settings. The topic is of significant relevance given the increasing global burden of neurodegenerative diseases.
Major Comments:
Abstract: The abstract provides a clear overview of the manuscript's content. However, it would be beneficial to include the main findings or conclusions drawn from the review to give readers a quick snapshot of the outcomes.
Introduction: The introduction sets the stage well by highlighting the importance of metformin and its potential role in neuroprotection. It might be beneficial to briefly touch upon the current challenges in treating neurodegenerative diseases to provide context.
Metformin Pharmacology: This section provides a detailed overview of metformin's pharmacokinetics and pharmacodynamics. It would be helpful to include a brief summary or a diagrammatic representation for readers less familiar with these concepts.
Mechanisms involved in glucoregulation: The authors have explained the role of AMPK in metformin's action. However, a more detailed discussion or a visual representation of the signaling pathways involved could enhance understanding.
AMPK signaling pathway: This section could benefit from a more structured presentation, possibly with subheadings or bullet points to delineate the different pathways and mechanisms.
Minor Comments:
There are some instances where references are clustered together (e.g., [4–6], [11–15]). It might be helpful to ensure that each reference is directly relevant to the statement it is supporting.
The manuscript mentions various neurodegenerative diseases, including Alzheimer's, Parkinson's, and Huntington's. It would be beneficial to have a separate section or subsection discussing the evidence for metformin's role in each of these diseases.
The authors might consider discussing any potential side effects or challenges associated with using metformin for neuroprotection. This would provide a more balanced view of its potential as a therapeutic agent.
Some sentences are quite lengthy and could be broken down for clarity.
Conclusion:
Overall, the manuscript is well-researched and presents a thorough analysis of the potential neuroprotective effects of metformin. With some structural and content modifications, it can serve as a valuable resource for researchers and clinicians in the field.
Author Response
Response to Reviewer 2:
First, we thank to editors and all the reviewers for their effort and the time spent to improve our work and for the insightful comments on our article. Below, we respond to each comment in an itemized fashion, hoping that the related changes to the manuscript are satisfactory, so that the paper can be accepted in its currently revised form.
The modifications made to the manuscript have been marked in the manuscript.
Here is a point-by-point response to the reviewers’ comments and concerns.
Comment 1: Abstract: The abstract provides a clear overview of the manuscript's content. However, it would be beneficial to include the main findings or conclusions drawn from the review to give readers a quick snapshot of the outcomes.
Thank you for the suggestion, we have accordingly modified the Abstract.
Metformin, a widely prescribed medication for type 2 diabetes, has garnered increasing attention for its potential neuroprotective properties due to the growing demand for treatments for Alzheimer's, Parkinson's, and motor neuron diseases. This review synthesizes experimental and clinical studies on metformin's mechanisms of action and potential therapeutic benefits for neurodegenerative disorders. A comprehensive search of electronic databases, including PubMed, MEDLINE, Embase, and Cochrane library, focuses on key phrases such as "metformin," "neuroprotection," and "neurodegenerative diseases," with data up to September 2023. Recent research on metformin's glucoregulatory mechanisms reveals new molecular targets, including the activation of the LKB1–AMPK signalling pathway, which is crucial for chronic administration of metformin. The pleiotropic impact may involve other stress kinases that are acutely activated. The precise role of respiratory chain complexes (I and IV), of the mitochondrial targets or of the lysosomes in metformin effects remains to be established by further research. Research on extrahepatic targets like the gut and microbiota, as well as its antioxidant and immunomodulatory properties, is crucial in understanding neurodegenerative dis-orders. Experimental data on animal models shows promising results, but clinical studies are inconclusive. Understanding the molecular targets and mechanisms of its effects could help design the clinical trials to explore and, hopefully, prove its therapeutic effect in neurodegenerative conditions.
Comment 2: Introduction: The introduction sets the stage well by highlighting the importance of metformin and its potential role in neuroprotection. It might be beneficial to briefly touch upon the current challenges in treating neurodegenerative diseases to provide context.
Thank you for the suggestions. We added the following paragraph in the introduction. Additionally, the references were updated.
Neurodegenerative diseases often require early detection and treatment, but current methods are limited. Current treatments focus on symptom management, but do not effectively slow disease progression [26,28]. The blood-brain barrier poses a challenge for drug delivery, and the heterogeneity of diseases complicates treatment approaches. The high cost of research and development further complicates the process, with clinical trials often failing to bring effective treatments to market [27]. Therefore, there is a need for dis-ease-modifying treatments to slow or stop disease progression [26].
Comment 3, 4, 5: Metformin Pharmacology: This section provides a detailed overview of metformin's pharmacokinetics and pharmacodynamics. It would be helpful to include a brief summary or a diagrammatic representation for readers less familiar with these concepts.
Mechanisms involved in glucoregulation: The authors have explained the role of AMPK in metformin's action. However, a more detailed discussion or a visual representation of the signaling pathways involved could enhance understanding.
AMPK signaling pathway: This section could benefit from a more structured presentation, possibly with subheadings or bullet points to delineate the different pathways and mechanisms.
Thank you for your suggestions. We agree on the need of clarifying the concepts and we have added four separate figures regarding metformin's pharmacokinetics, mechanisms involved in glucoregulation, AMPK signaling pathways and signaling pathways involved in neuroprotection:
Figure 1. Overview of transporters relevant to metformin pharmacokinetics. MATE multidrug and toxin extrusion transporter, OCT organic cation transporter, PMAT plasma membrane monoamine transporter, SERT serotonin transporter, THTR2 thiamine transporter.
Parts of the figure were drawn by using pictures from Servier Medical Art. Servier Medical Art by Servier is licensed under a Creative Commons Attribution 3.0 Unported License (https://creativecommons.org/licenses/by/3.0/)
Figure 2. AMPK signaling pathway. AMPK AMP-activated protein kinase, LKB1 Liver kinase B1.
Figure 3. Proposed mechanism of action for metformin Left: Low dose of metformin interferes with the vacuolar H+-ATPase in the lysosomal proton pump (v-ATPase). The interaction between metformin and PEN2 (presenilin enhancer protein 2) forms a complex involving ATP6AP1. The AMPK-dependent model of metformin-induced inhibition of gluconeogenesis is based on the activation of the LKB1–AMPK signalling pathway. Middle: Metformin causes a slight inhibition of mitochondrial respiratory chain complex I in the liver, which results in a moderate reduction in ATP production and an increase in AMP (inhibiting Fruc-tose-1,6-Bisphosphatase Deficiency FBP1 and adenylate cyclase AC) levels in the cells. AMP-activated protein kinase (AMPK) is activated by the metformin-induced rise in the AMP to ATP ratio and increases GLP1 and inactivates/inhibits G6Pase(glucose-6-phosphatase), PEPCK (phosphoenolpyruvate carbox-ykinase) and HMG-CoA (3-hydroxy-3-methylglutaryl coenzyme A). The inhibition of complex I by met-formin is also accompanied by an increased ratio of lactate/pyruvate, and reduced glucose synthesis. Right: Mitochondrial glycerol-3-phosphate dehydrogenase (mGPDH)-dependent. Because metformin directly in-hibits mGPDH, there is a decrease in lactate gluconeogenesis, a reduction in the activity of the glycerol–phosphate shuttle (which moves NADH from the cytosol to the mitochondria), and a rise in the cytosolic redox state (NADH:NAD+).
Parts of the figure were drawn by using pictures from Servier Medical Art. Servier Medical Art by Servier is licensed under a Creative Commons Attribution 3.0 Unported License (https://creativecommons.org/licenses/by/3.0/)
Figure 4. Signaling pathways involved in neuroprotective mechanisms. MAPK mitogen-activated protein kinase, Pi3K phosphatidylinositol 3-kinase, FOXO3 forkhead box O 3, ROS reactive oxygen species, TBARS thiobarbituric acid reactive substances, MDA malondialdehyde, GSSG oxidized form of glutathione, GSH reduced form of glutathione, AMPK adenosine monophosphate-activated protein kinase, TNF-α tumor Necrosis Factor-α, IL-1β interleukin 1 beta, IL-6 interleukin 6.
Comment6: There are some instances where references are clustered together (e.g., [4–6], [11–15]). It might be helpful to ensure that each reference is directly relevant to the statement it is supporting.
We thank the reviewer for the observation, we have changed the clustered references.
Comment 7: The manuscript mentions various neurodegenerative diseases, including Alzheimer's, Parkinson's, and Huntington's. It would be beneficial to have a separate section or subsection discussing the evidence for metformin's role in each of these diseases.
Thank you for the valuable suggestion. We have re-structured our review according to the observation and you can find a separate section for Alzheimer’s disease at page 16, Parkinson's at page 20, Huntington's at page 22, Epilepsy at page 23 and Fragile X syndrome at page 24.
Comment 8: The authors might consider discussing any potential side effects or challenges associated with using metformin for neuroprotection. This would provide a more balanced view of its potential as a therapeutic agent.
Thank you for your suggestions. Some sections (Alzheimer’s’ disease, Parkinson's, Ameliorative effect of cognitive impairment and memory loss) include studies that demonstrated contrary results. Further clinical trials are needed to understand its potential benefits and appropriate application.
The potential side effects associated with metformin are described in the paragraphs no 13, 14,15,16 in the section dedicated to Alzheimer’s disease and in the paragraphs no 6,7,8,9 in the section dedicated to Parkinson’s disease.
Comment 9: Some sentences are quite lengthy and could be broken down for clarity.
Thank you for the observation, we have thoroughly revised the manuscript and have made changes to improve clarity.

Reviewer 3 Report
Comments and Suggestions for Authors
In the current review, the authors showed relevant information about the neuroprotective effects of Metformin considering in vivo and in vitro models of neurodegenerative disease. However, the text presentation and organization should be improved for future publishing.
Minor comments are highlighted in the attached file.
Major comments
- The authors should consider adding more information about metformin transporters and receptors in the nervous system.
- The authors should consider presenting the discussed item showing the in vivo and in vitro literature results in separate items.
- The authors could create a schematic figure summarizes the main metformin mechanism in T2D and metabolic conditions, showing receptors, transporters, and pathways, and a figure showing the neuroprotective pathways and proprieties in the nervous system, especially in the brain related to memory and cognitive impairments.

Author Response
Response to Reviewer 3:
First, we thank to editors and all the reviewers for their effort and the time spent to improve our work and for the insightful comments on our article. Below, we respond to each comment in an itemized fashion, hoping that the related changes to the manuscript are satisfactory, so that the paper can be accepted in its currently revised form.
The modifications made to the manuscript have been marked in the manuscript.
Here is a point-by-point response to the reviewers’ comments and concerns.
Comment 1: The authors should consider adding more information about metformin transporters and receptors in the nervous system.
Thank you for suggestion. We added the following paragraph in the introduction. Additionally, the references were updated.
Although the precise mechanisms by which metformin is transported in the central nervous system are still unclear, recent in vivo and in vitro studies showed that it can pass the blood-brain barrier and can work by activating certain neurons and neuroglia to pro-vide its desired effects [53,54]. Moreover, metformin transporting has been linked to the equilibrative nucleoside transporter (ENT) family, of which PMAT is a member, and which is present in many bodily tissues, including the brain and central nervous system. [44]. Blood-brain barrier and brain express OCT3, as well [55]. Research on the metformin transporter in the neurological system is currently ongoing, and further investigation is required to clarify the precise processes at play. Nevertheless, the fact that metformin may penetrate the blood-brain barrier and operate neurologically points to possible therapeutic benefits for the central nervous system [53,54].
Comment 2: The authors should consider presenting the discussed item showing the in vivo and in vitro literature results in separate items.
Thank you for the observation and suggestion. Generally, we have structured each of the sections on Alzheimer’s’ disease, Parkinson's, Huntington's, Epilepsy, and Fragile X syndrome presenting data derived from in vitro studies, animal research, and any type of existing clinical studies (in consecutive, different paragraphs). However, as in some cases the same idea/concept might have been supported by data derived from both in-vivo/in-vitro studies, we did not completely separate those. Following the main focus of our article and the reviewer observation, we considered that changing the title of the manuscript into “Metformin: The winding path from understanding its molecular mechanisms to proving therapeutic benefits in neurodegenerative disorders” would better reflect its content.
Comment 3: - The authors could create a schematic figure summarizes the main metformin mechanism in T2D and metabolic conditions, showing receptors, transporters, and pathways, and a figure showing the neuroprotective pathways and proprieties in the nervous system, especially in the brain related to memory and cognitive impairments.
Thank you for your suggestions. We agree on the need of clarifying the concepts and we have added four separate figures regarding metformin's pharmacokinetics, mechanisms involved in glucoregulation, AMPK signaling pathways and signaling pathways involved in neuroprotection:
Thank you for your suggestions. We agree on the need of clarifying the concepts and we have added four separate figures regarding metformin's pharmacokinetics, mechanisms involved in glucoregulation, AMPK signaling pathways and signaling pathways involved in neuroprotection:
Figure 1. Overview of transporters relevant to metformin pharmacokinetics. MATE multidrug and toxin extrusion transporter, OCT organic cation transporter, PMAT plasma membrane monoamine transporter, SERT serotonin transporter, THTR2 thiamine transporter.
Parts of the figure were drawn by using pictures from Servier Medical Art. Servier Medical Art by Servier is licensed under a Creative Commons Attribution 3.0 Unported License (https://creativecommons.org/licenses/by/3.0/)
Figure 2. AMPK signaling pathway. AMPK AMP-activated protein kinase, LKB1 Liver kinase B1.
Figure 3. Proposed mechanism of action for metformin Left: Low dose of metformin interferes with the vacuolar H+-ATPase in the lysosomal proton pump (v-ATPase). The interaction between metformin and PEN2 (presenilin enhancer protein 2) forms a complex involving ATP6AP1. The AMPK-dependent model of metformin-induced inhibition of gluconeogenesis is based on the activation of the LKB1–AMPK signalling pathway. Middle: Metformin causes a slight inhibition of mitochondrial respiratory chain complex I in the liver, which results in a moderate reduction in ATP production and an increase in AMP (inhibiting Fruc-tose-1,6-Bisphosphatase Deficiency FBP1 and adenylate cyclase AC) levels in the cells. AMP-activated protein kinase (AMPK) is activated by the metformin-induced rise in the AMP to ATP ratio and increases GLP1 and inactivates/inhibits G6Pase(glucose-6-phosphatase), PEPCK (phosphoenolpyruvate carbox-ykinase) and HMG-CoA (3-hydroxy-3-methylglutaryl coenzyme A). The inhibition of complex I by met-formin is also accompanied by an increased ratio of lactate/pyruvate, and reduced glucose synthesis. Right: Mitochondrial glycerol-3-phosphate dehydrogenase (mGPDH)-dependent. Because metformin directly in-hibits mGPDH, there is a decrease in lactate gluconeogenesis, a reduction in the activity of the glycerol–phosphate shuttle (which moves NADH from the cytosol to the mitochondria), and a rise in the cytosolic redox state (NADH:NAD+).
Parts of the figure were drawn by using pictures from Servier Medical Art. Servier Medical Art by Servier is licensed under a Creative Commons Attribution 3.0 Unported License (https://creativecommons.org/licenses/by/3.0/)
Figure 4. Signaling pathways involved in neuroprotective mechanisms. MAPK mitogen-activated protein kinase, Pi3K phosphatidylinositol 3-kinase, FOXO3 forkhead box O 3, ROS reactive oxygen species, TBARS thiobarbituric acid reactive substances, MDA malondialdehyde, GSSG oxidized form of glutathione, GSH reduced form of glutathione, AMPK adenosine monophosphate-activated protein kinase, TNF-α tumor Necrosis Factor-α, IL-1β interleukin 1 beta, IL-6 interleukin 6.
Comment 4: Minor comments are highlighted in the attached file.
Thank you for your careful review of the article and for your comment. We reviewed the entire manuscript for typographical errors and made the needed corrections.

Reviewer 4 Report
Comments and Suggestions for Authors
This research articles is interesting and well presented, and provides useful information about Narrowing the gap between in vitro and in vivo studies re- 2 grading metformin's neuroprotective effects.
The manuscript is written well and acceptable for publication but need detail revision and correction before publication.
1) In the abstract the author mentioned the Google Scholar as Data base, but in fact it is search engine. Need correction
2) The Author ignored the most important Cochran Data base for systematic review. I think he should focus more on this data base for systematic review.
3) Also the author mentioned good effect of Metformin against Alzheimer but no such effect in animal model? Why
4) Manuscript language can be improved as it needs minor correction, spelling, spacing etc.
5) Endnote should be used for citation and reference if not used.

This research articles is interesting and well presented, and provides useful information about Narrowing the gap between in vitro and in vivo studies re- 2 grading metformin's neuroprotective effects.
The manuscript is written well and acceptable for publication but need detail revision and correction before publication.
1) In the abstract the author mentioned the Google Scholar as Data base, but in fact it is search engine. Need correction
2) The Author ignored the most important Cochran Data base for systematic review. I think he should focus more on this data base for systematic review.
3) Also the author mentioned good effect of Metformin against Alzheimer but no such effect in animal model? Why
4) Manuscript language can be improved as it needs minor correction, spelling, spacing etc.
5) Endnote should be used for citation and reference if not used.
Author Response
Response to Reviewer 4:
First, we thank to editors and all the reviewers for their effort and the time spent to improve our work and for the insightful comments on our article. Below, we respond to each comment in an itemized fashion, hoping that the related changes to the manuscript are satisfactory, so that the paper can be accepted in its currently revised form.
The modifications made to the manuscript have been marked in the manuscript.
Here is a point-by-point response to the reviewers’ comments and concerns.
Comment 1: In the abstract the author mentioned the Google Scholar as Data base, but in fact it is search engine. Need correction.
Thank you for your careful review of the article and for your comment, we have deleted Google Scholar.
Comment 2: The Author ignored the most important Cochran Data base for systematic review. I think he should focus more on this data base for systematic review.
Thank you for the observation. We agree on that, it was an error not to mention it and we corrected it. It fact, we performed the searches in Cochrane, but, unfortunately, only one meta-analysis regarding the neuroprotective effect of metformin was found: Areosa Sastre A, Vernooij RWM, González‐Colaço Harmand M, Martínez G. Effect of the treatment of Type 2 diabetes mellitus on the development of cognitive impairment and dementia. Cochrane Database of Systematic Reviews 2017, Issue 6. Art. No.: CD003804. DOI: 10.1002/14651858.CD003804.pub2. (reference no 178 in our manuscript)
Comment 3: Also the author mentioned good effect of Metformin against Alzheimer but no such effect in animal model? Why
We are grateful for this comment. The effect against Alzheimer in animal model is described in the paragraphs no 4, 5, 6 in the section dedicated to Alzheimer’s disease.
Comment 4: Manuscript language can be improved as it needs minor correction, spelling, spacing etc.
Thank you for your careful review of the article and for your comment. We reviewed the entire manuscript for typographical errors and made the required corrections.
Comment 5: Endnote should be used for citation and reference if not used.
Thank you for your suggestion. We used Mendeley Reference Manager while writing the manuscript.

Round 2
Reviewer 3 Report
Comments and Suggestions for Authors
The authors reviewed the manuscript and presented an updated version. They improved the discussion section and added figures to summarize the metformin pathways. However, there is a missed link with neurodegenerative disease. I suggest a figure illustrating a neurodegenerative disease condition and showing the transporters and pathways are activate in CNS to promote terapeutics benefits. In line with this, the text could be improve a focus on neurodegeneration process.
Please review red highlights on the file for minor test corrections.
The figures resolution could be improve.

Author Response
Dear Ms/Mrs,
We are grateful for the opportunity to submit a revised version of our article, titled "Narrowing the gap between in vitro and in vivo studies regarding metformin's neuroprotective effects." We appreciate the time and work you and the reviewers put into providing us with informative feedback on our post. We appreciate the reviewers' insightful feedback on our work. All of the reviewers' recommendations have been included into the changes we have made. The changes made to the manuscript have been highlighted.
Here is a point-by-point response to the reviewers’ comments and concerns.
Comments from Reviewer 3:
Comment 1: The authors reviewed the manuscript and presented an updated version. They improved the discussion section and added figures to summarize the metformin pathways. However, there is a missed link with neurodegenerative disease. I suggest a figure illustrating a neurodegenerative disease condition and showing the transporters and pathways are activate in CNS to promote therapeutic benefits. In line with this, the text could be improving a focus on neurodegeneration process.
We appreciate your insightful comment and suggestion for further emphasizing the link with neurodegenerative diseases in our manuscript. In response to your recommendation, we have included a new diagram depicting the neurodegenerative process in chapter 3. This addition aims to provide a visual representation that enhances the understanding of the link between our research and neurodegenerative diseases.
To complement this, we have revised Figure 4 to ensure a clearer and more comprehensive explanation of the neuroprotection mechanism.
Moreover, we added the following paragraphs to enhance the clarity and context of our manuscript. Additionally, the references were updated.
- Mechanisms involved in central nervous system functions and neuroprotection
The complex regulation of the central nervous system's activities in the context of neurodegeneration is vulnerable to a variety of complex processes, each of which contributes in a different but related way to the development of crippling disorders. The mechanisms that are involved—such as inflammation, oxidative damage, mitochondrial dysfunction, impaired autophagy, aberrant protein phosphorylation, and disruption of insulin signaling and glucose homeostasis —become crucial factors in determining the course of neuronal health within this complex web. Figure 4 provides an outline of the mechanisms involved in neurodegeneration.
Molecular events play a significant role in neurodegenerative disorders, marked by an escalation of inflammatory cytokines orchestrated by microglia and astrocytes, inducing a pro-inflammatory state in the central nervous system. Elevated levels of ROS give rise to oxidative stress, disrupting the delicate balance of redox homeostasis. NF-κB signaling amplifies the inflammatory response, fostering the progression of neurodegeneration. The decline in antioxidants, crucial guardians against oxidative stress, heightens vulnerability. Furthermore, alterations in insulin signaling and glucose homeostasis contribute to metabolic dysregulation, a critical factor in neurodegenerative processes. Additionally, aging shares almost parallel mechanisms with neurodegeneration. The impact of metformin across these pathways suggests its potential to mitigate these intricate events.
- The winding path from neuroprotective mechanisms to proving benefits in neurodegenerative diseases:
The signaling pathways which can interfere with neurodegeneration are crucial to deciphering the complexities of neuroprotection and translate these discoveries into meaningful advances for individuals affected by neurodegenerative diseases. Long-term reactive oxygen species (ROS) produced in mitochondria during oxidative phosphorylation have been thought to be one of the main causes of aging-related DNA damage. Another process involves the constitutive activation of mTOR/S6 signaling, which is a sensor of nutrients and mitogens [161]. Aging inhibition affects the cellular pathways upstream of mTOR, including the PI3K, AKT, MAPK, and IGF-1/GH axis [162]. Inhibition IGF-1/GH axis reduces metabolic rate, cell proliferation, oxidative stress, decreases the accumulation of senescent cells and counteractes inflammaging [163]. mTOR signaling has been associated to accelerated aging, and dysregulation of mTOR signaling has also been connected to the advancement of cancer, inflammatory and neurological illnesses, as well as T2DM [164].. Since mTOR is indirectly inhibited by AMPK activation, metformin, an AMPK activator, has been demonstrated to have gero-suppressive effects [165].
Cognitive dysfunction, which encompasses conditions like delirium, mild cognitive deficits, and dementia, signifies a significant deterioration from an individual's previously achieved cognitive functioning level [166,167]. Some epidemiological studies have shown that metformin reduces the incidence of a variety of age-related illnesses as well as overall mortality. Importantly, this phenomenon was found in both diabetic and non-diabetic subjects [159,168]. A randomized, double-blind, placebo-controlled crossover study found that metformin had both metabolic and non-metabolic effects linked with aging in the elderly, indicating that metformin has an anti-aging impact [169]. Metformin has lately been proposed as an anti-aging medicine due to its widespread usage in clinical practice, well-known pharmacokinetics, and low toxicity.
Comment 2: Please review the red highlights on the file for minor test corrections.
Thank you for your careful review of the article and for your comment. We reviewed the entire manuscript for typographical errors.
Comment 3: The figures resolution could be improved.
Thank you for your valuable feedback. Regarding your comment on the figures' resolution, we acknowledge the importance of clear and high-quality visuals to enhance the understanding of the presented data. We addressed this concern by re-examining the resolution of the figures and replacing them with higher-resolution versions.

Round 3
Reviewer 3 Report
Comments and Suggestions for Authors
The authors accepted my last recommendations and improved the study's quality presentation. Please see the highlighted text on the attached file for minor corrections. Do a careful review before resubmitting!

Author Response
Thank you for your diligent review and constructive feedback. We appreciate your acknowledgment of the efforts we've made to incorporate your previous recommendations and enhance the overall quality of the article's presentation. We have carefully examined the highlighted text in the attached file and have made the necessary minor corrections.
We conducted a thorough review of the entire manuscript before resubmitting it. We are grateful for your time and expertise, and we believe that your input will contribute significantly to the refinement of our work.